# FairDD: Fair Dataset Distillation

**Qihang Zhou**[*], **Shenhao Fang**[*], **Shibo He**[†], **Wenchao Meng, Jiming Chen**
Zhejiang University
{zqhang, 22460454, s18he, wmengzju, cjm}@zju.edu.cn

## Abstract

Condensing large datasets into smaller synthetic counterparts has demonstrated its promise for image classification. However, previous research has overlooked a crucial concern in image recognition: ensuring that models trained on condensed datasets are unbiased towards protected attributes (PA), such as gender and race. Our investigation reveals that dataset distillation fails to alleviate the unfairness towards minority groups within original datasets. Moreover, this bias typically worsens in the condensed datasets due to their smaller size. To bridge the research gap, we propose a novel fair dataset distillation (FDD) framework, namely FairDD, which can be seamlessly applied to diverse matching-based DD approaches (DDs), requiring no modifications to their original architectures. The key innovation of FairDD lies in synchronously matching synthetic datasets to PA-wise groups of original datasets, rather than indiscriminate alignment to the whole distributions in vanilla DDs, dominated by majority groups. This synchronized matching allows synthetic datasets to avoid collapsing into majority groups and bootstrap their balanced generation to all PA groups. Consequently, FairDD could effectively regularize vanilla DDs to favor biased generation toward minority groups while maintaining the accuracy of target attributes. Theoretical analyses and extensive experimental evaluations demonstrate that FairDD significantly improves fairness compared to vanilla DDs, with a promising trade-off between fairness and accuracy. Its consistent superiority across diverse DDs, spanning Distribution and Gradient Matching, establishes it as a versatile FDD approach. Code is available at `https://github.com/zqhang/FairDD`.

## 1 Introduction

Deep learning has witnessed remarkable success in computer vision, particularly with recent breakthroughs in vision models [45, 28, 47, 33, 72]. Their vision backbones, such as ResNet [20] and ViT [16], are data-hungry models that require extensive amounts of data for optimization. Dataset Distillation (DD) [60, 67, 69, 6, 58, 32, 12, 38, 18, 22, 7, 8, 71] provides a promising solution to alleviate this data requirement by condensing the original large dataset into more informative and smaller counterparts [42, 10]. Despite its appeal, existing researches focus on ensuring that models trained on condensed datasets perform comparable accuracy to those trained on the original dataset in terms of target attributes (TA) [13, 40, 56]. However, they have overlooked enabling the fairness of trained models with respect to protected attributes (PA).

Unfairness typically arises from imbalanced sample distributions among PA in the empirical training datasets. When the original datasets suffer from the PA imbalance, the corresponding datasets condensed by vanilla DDs inherit and amplify this bias in Fig. 1(e). Since vanilla DDs tend to cover TA distribution for image classification, and as a result, it naturally leads to more synthetic samples located in majority groups compared to minority groups w.r.t. PA, as shown in Figs. 1(a), 1(b), 1(c), and 1(d). In this case, these condensed datasets retain the imbalance between protected attributes,

---

[*]Equal contribution. † Corresponding authors.

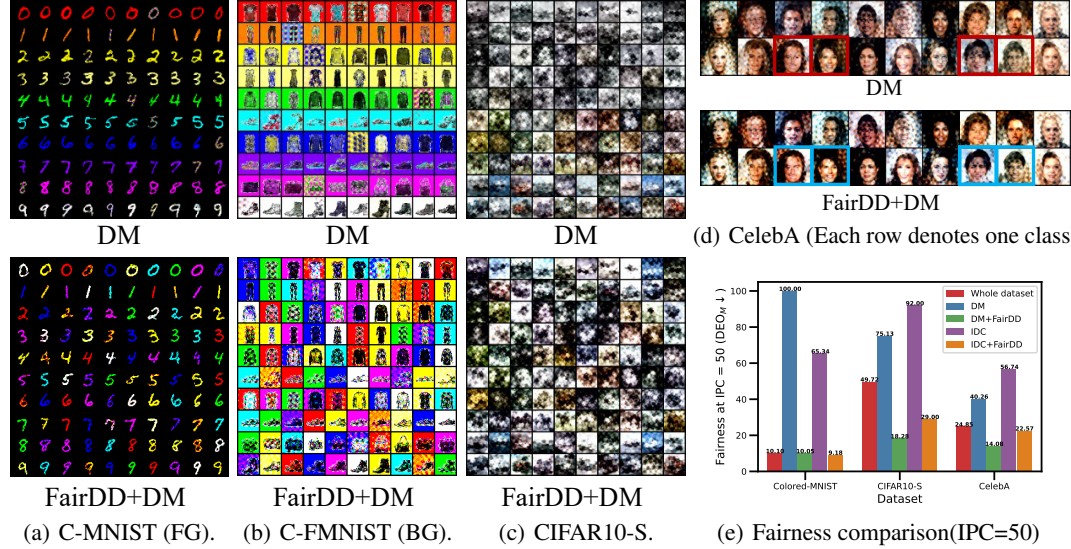

(a) C-MNIST (FG).    (b) C-FMNIST (BG).    (c) CIFAR10-S.    (e) Fairness comparison(IPC=50)

Figure 1: Visualization comparison on $\mathcal{S}$ at IPC = 10 for diverse datasets. FairDD successfully mitigates the bias from original datasets in (a) foreground digital color, (b) background color, (c) foreground and background grayscale (d) real-world bias. (e) vanilla DDs exacerbate the unfairness.

thereby rendering the model trained on them unfair. Moreover, the reduced size of the condensed datasets typically amplifies the bias present in the original datasets, especially when there is a significant gap in size between the original and condensed datasets, such as image per class (IPC) 1000 vs. 10. Therefore, it is worthwhile to broaden the scope of DDs to encompass both TA accuracy and PA fairness. Recent works [14, 41] attempt to address the class/TA-level long-tailed phenomenon [71] and spurious correlations [13] to improve the classification performance [56, 59], but the exploration on visual fairness is still blank.

To bridge this gap, we propose FairDD, a novel FDD framework that achieves PA fairness in models trained on condensed datasets, even when the original data exhibit PA imbalance. Note that FDD addresses data-level fairness: how to generate a distilled dataset that is inherently unbiased, agnostic to the downstream model. Instead, traditional fairness literatures focus on model-level fairness: how to train a fair model given a dataset. These two perspectives approach fairness from distinct yet complementary directions. FDD requires simultaneously maintaining TA accuracy and improving PA fairness. It is challenging, as the algorithm must properly balance the emphasis across all groups — reducing the dominance of majority groups while maintaining their TA distributional coverage, and preserving minority groups to mitigate PA bias.

FairDD tackles this challenge by **(1)** partitioning the empirical training distribution into different groups according to PA and decomposing the single alignment target of vanilla DDs into PA-wise sub-targets. **(2)** synchronously matching synthetic samples to these PA groups, which equally bootstraps synthetic datasets to each PA group without involving the specific group size. In doing so, we reformulated the optimization objectives of vanilla DDs into fairDD-style versions. This allows FairDD to mitigate the effect of imbalanced PA on the generation of $\mathcal{S}$ and prevents $\mathcal{S}$ from collapsing into the majority group. In Fig. 1(d), FairDD synthesizes more male samples (highlighted by blue squares) within an attributive class originally dominated by females. Meanwhile, FairDD could also achieve the comprehensive coverage of the entire distribution for TA accuracy. We provide a theoretical guarantee that FairDD could improve PA fairness while maintaining TA accuracy.

Extensive experiments demonstrate that our framework effectively mitigates the unfairness in datasets of highly diverse bias. FairDD substantially improves data fairness trained on condensed datasets compared to various vanilla DDs. FairDD demonstrates its versatility across diverse DDs, including Distribution Matching (DM) and Gradient Matching (GM) [2]. Our main contributions are as follows:

---

[2]We do not apply FairDD to Trajectory Matching (TM) because it would require additional model trajectories trained on minority groups, prone to overfitting due to their limited sample sizes.

- To the best of our knowledge, our research is the first attempt to incorporate visual fairness into DDs explicitly. We reveal that vanilla DDs fail to mitigate the bias in original datasets and may exacerbate it due to the limited synthetic samples, leading to severe PA bias in the model trained by the resulting condensed dataset.

- We introduce a novel FDD framework called FairDD, which proposes synchronized matching to align synthetic samples to all PA groups partitioned from the original data distribution. This allows the generated synthetic samples to be agnostic to PA imbalance of original datasets while maintaining the overall distributional coverage of TA.

- Extensive empirical experiments demonstrate that FairDD is a generalist to significantly mitigate the unfairness of vanilla DDs. Its consistent superiority is observed across various DDs, including DM and GM.

## 2   Related Work

**Dataset distillation**   Dataset distillation has been broadly applied to many important fields [30, 21, 17, 9]. The first work [60] attempts to formulate dataset distillation as a bi-level optimization problem. However, the two folds of the optimization process are time-consuming. Neural tangent kernel [23] is utilized to obtain the closed form of the inner loop [44, 37, 73]. Some previous works propose surrogate objectives to achieve comparable even better performance, including matching-based methods like GM [70, 67, 31], DM [69, 58, 68], TM [6, 12], soft label learning [4, 54, 62, 52], and factorization [27, 15, 35, 31]. Recent works [14, 41] attempt to mitigate bias to improve classification accuracy on the TA without considering protected attributes (PA), staying within the traditional setting of DD [13, 71, 56, 59]. The work [13] mitigates sample-wise bias by assigning higher weights to samples located in low-density regions of the original data distribution, while they neglect fairness concerning PA. It fails to guarantee that the alignment objective is unbiased across all attribute groups, nor does it ensure adequate distribution coverage. Instead, our methods have a fairness alignment objective to facilitate unbiased data distillation; In addition, we provide a theoretical proof to guarantee the distribution coverage for TA. This makes FairDD with a good balance between fairness and accuracy. We provide a performance comparison in Appendix E.

**Visual fairness**   Current literature on fairness aims to train a model that outputs fair logits under class-imbalanced datasets [5]. According to the stage of bias mitigation, the research field of fairness algorithm [3] can be classified into three branches: Pre-processing [11, 39, 46, 51], In-processing [1, 24, 64, 66, 26, 61, 25, 65], and Post-processing [2, 19]. They learn fair representations without involving information condensation [53, 55, 57, 48]. Fairness-aware synthetic data generation serves as a pre-processing for fairness. They frame fairness mitigation as a data-to-data translation problem, and utilize generative models [63] to produce fairer datasets with respect to protected groups [46, 50]. However, they do not consider the aspect of information condensation. Instead, our work aims to reduce bias in condensed datasets by: (1) ensuring that the information from the original datasets is effectively distilled into the condensed datasets, and (2) simultaneously mitigating both the inherent bias of the original dataset and the bias exacerbated by vanilla dataset condensation. Once the data is condensed, we can train a fair model without any further human intervention.

## 3   Preliminaries

**Dataset Distillation.**   Given a vast dataset $\mathcal{T} = \{(x_i, y_i)\}_{i=1}^{N}$, DDs aim to condense original dataset $\mathcal{T}$ into a smaller dataset $\mathcal{S} = \{(x_i, y_i)\}_{i=1}^{M}$ via distillation algorithm $\mathrm{Alg}$ with nerual networks, parameterized by $\theta$. Randomly initialized classification network $g_\psi$ should maintain the same empirical risk whether it is trained on $\mathcal{S}$ or $\mathcal{T}$.

$$\mathcal{S}^* = \underset{S}{\arg\min}\, \mathrm{Alg}(\mathcal{S}, \mathcal{T}, \theta), \quad \mathbb{E}_{\psi \sim \Psi}[\ell(g_\psi; \mathcal{S})] \simeq \mathbb{E}_{\psi \sim \Psi}[\ell(g_\psi; \mathcal{T})],$$

where $\Psi$ and $\ell(\cdot)$ represent the parameter space and loss function, respectively. The pioneering work [60] formulates $\mathrm{Alg}$ as a bi-level optimization problem. However, such an optimization process is time-consuming and unstable. Recent works circumvent it and propose surrogate matching objectives to achieve comparable and even better performance. This research line is collectively referred to as the DMF, and our paper primarily studies one-stage GM [70, 67] and DM [69, 58, 68]. We leave it for future exploration.

**Visual Fairness**   Visual fairness is an important field to mitigate discrimination against minority groups. Group fairness requires no statistical disparity to different groups in terms of PA, such as race and gender. This means that an ideal fair model should make independent predictions between TA and PA. One of the common fairness criteria is equalized odds (EO), which computes the prediction accuracy of PA conditioned on TA, to evaluate the level of conditional independence between PA and TA. We use two types of difference of equalized odds $\text{DEO}_\text{M}$ and $\text{DEO}_\text{A}$ from the worst and averaged levels. Formally, given the PA set $\mathcal{A} = \{a_1, a_2, ..., a_p\}$, where $p$ denotes the number of protected attributes. $\text{DEO}_\text{M}$ and $\text{DEO}_\text{A}$ [26] can be formulated mathematically as follows:

$$\text{DEO}_\text{M} = \max_{y \in \mathcal{Y}} \max_{a_i, a_j \in \mathcal{A} \& a_i \neq a_j} \left| P(\hat{Y} = y | Y = y, A = a_i) - P(\hat{Y} = y | Y = y, A = a_j) \right|,$$

$$\text{DEO}_\text{A} = \operatorname*{mean}_{y \in \mathcal{Y}} \max_{a_i, a_j \in \mathcal{A} \& a_i \neq a_j} \left| P(\hat{Y} = y | Y = y, A = a_i) - P(\hat{Y} = y | Y = y, A = a_j) \right|.$$

# 4   A Close Look at Dataset Distillation

**A unified perspective for Data Match Framework.**   The essence of the DMF lies in choosing the target signs of original samples that effectively represent their characteristics for image recognition, and then aligning these signals as a proxy task to optimize the condensed dataset. The target signal $\phi(x; \theta)$ is typically the key information from feature extraction or optimization process using a randomly initialized network parameterized by $\theta$. For example, GM aligns the gradient information produced by $\mathcal{T}$ with that of the condensed $\mathcal{S}$. Instead, DM matches the embedding distributions of $\mathcal{T}$ and $\mathcal{S}$. As for these approaches in DMF. we can unify the optimization objective as $\mathcal{L}(\mathcal{S}; \theta, \mathcal{T})$:

$$\mathcal{L}(\mathcal{S}; \theta, \mathcal{T}) := \sum_{y \in \mathcal{Y}} \mathcal{D}\left( \mathbb{E}[\phi_{x \sim \mathcal{T}_y}(x; \theta)], \mathbb{E}[\phi_{x \sim \mathcal{S}_y}(x; \theta)] \right), \tag{1}$$

where $\mathbb{E}[\phi_{x \sim \mathcal{T}_y}(x; \theta)] \in \mathbb{R}^C$ and $\mathbb{E}[\phi_{x \sim \mathcal{S}_y}(x; \theta)] \in \mathbb{R}^C$ are represented expectation vectors of the target signs on $\mathcal{T}$ and $\mathcal{S}$, respectively. $\mathcal{D}(\cdot, \cdot)$ is a distance function. In DMF, MSE is adopted in DM and DREAM, and MAE is used in IDC.

**Why do vanilla DDs fail to mitigate PA imbalance?**   Given the dataset $\mathcal{T} = \{(x_i, y_i, a_i)\}_{i=1}^N$, $a_i \in \mathcal{A}$, let us define the class-level sample ratio $\mathcal{R}_y = \{r_y^{a_1}, r_y^{a_2}, ..., r_y^{a_p}\}$, where $r_y^{a_i} = |\mathcal{T}_y^{a_i}| / |\mathcal{T}_y|$, and $| \cdot |$ represents the cardinal number of a set. Current DDs' paradigms focus on preserving TA representativeness for image recognition. Here, we decompose the whole expectation into the expectation of PA-wise groups, i.e, $\mathbb{E}[\phi_{x \sim \mathcal{T}_y}(x; \theta)] = \sum_{a_i \in \mathcal{A}} r_y^{a_i} \mathbb{E}[\phi_{x \sim \mathcal{T}_y^{a_i}}(x; \theta)]$, and thus Eq. 1 can be rewritten as follows:

$$\mathcal{L}(\mathcal{S}; \theta, \mathcal{T}) := \sum_{y \in \mathcal{Y}} \mathcal{D}\left( \sum_{a_i \in \mathcal{A}} r_y^{a_i} \mathbb{E}[\phi_{x \sim \mathcal{T}_y^{a_i}}(x; \theta)], \mathbb{E}[\phi_{x \sim \mathcal{S}_y}(x; \theta)] \right). \tag{2}$$

From Eq. 2, the optimization objective of class $y$ is weighted by the sample ratio $r_y^{a_i}$ from different groups. When $\mathcal{T}$ suffers from PA imbalance, e.g., $r_y^{a_j} \gg \sum_{i \neq j} r_y^{a_i}$, the majority group indexed by $i$ contributes more to the alignment compared to minority groups. In other words, $\mathcal{S}$ tends to produce more samples belonging to group $i$ for the total loss minimization. The objective of vanilla DDs suffers from PA imbalance within $\mathcal{T}$.

Next, we further study how the resulting $\mathcal{S}$ is affected by sample ratio $r_y^{a_i}$ of different groups. To this end, we assume that the optimization process could reach the optimal solution for each class, and as a result, the final resulting $\mathcal{S}$ satisfies the condition that the derivative of the objective function with respect to $\mathbb{E}[\phi_{x \sim \mathcal{S}_y}(x; \theta)]$ equals 0, i.e., $\frac{\partial \mathcal{D}(\sum_{a_i \in \mathcal{A}} r_y^{a_i} \mathbb{E}[\phi_{x \sim \mathcal{T}_y^{a_i}}(x; \theta)], \mathbb{E}[\phi_{x \sim \mathcal{S}_y}(x; \theta)])}{\partial \mathbb{E}[\phi_{x \sim \mathcal{S}_y}(x; \theta)]} = 0$. Now, let's delve into the specific distance metrics used in vanilla DDs, where the most commonly used metrics are MAE, MSE, and cosine distance. We could compute the optimal point of $\mathbb{E}[\phi_{x \sim \mathcal{S}_y}(x; \theta)]$ could reach under these metrics:

$$\mathbb{E}[\phi_{x \sim \mathcal{S}_y}(x; \theta)] = \lambda \sum_{a_i \in \mathcal{A}} r_y^{a_i} \mathbb{E}[\phi_{x \sim \mathcal{T}_y^{a_i}}(x; \theta)], \tag{3}$$

Where $\lambda$ is a constant shared across all groups, equal to 1 for MAE and MSE, and equal to $\frac{\| \mathbb{E}[\phi_{x \sim \mathcal{S}_y}(x; \theta)] \|_2}{\| \sum_{a_i \in \mathcal{A}} r_y^{a_i} \mathbb{E}[\phi_{x \sim \mathcal{T}_y^{a_i}}(x; \theta)] \|_2}$ for the cosine loss. Eq. 3 presents that the expectation of synthetic samples $\mathbb{E}[\phi_{x \sim \mathcal{S}_y}(x; \theta)]$ ultimately converges to an average on expectations of all PA groups, weighted by their respective sample ratios $r_y^{a_i}$. This indicates that vanilla DDs naturally favor majority groups, causing $\mathcal{S}$ to shift towards them and inherit their biases.

When original datasets suffer from PA imbalance, e.g., $r_y^{a_j} \gg \sum_{i \neq j} r_y^{a_i}$, the unfairness of the synthetic dataset stems from two different aspects: 1) **The majority group renders synthetic samples to locate its region from Eq. 3.** 2) According to Eq. 2, the large sample quantities of the majority group contribute more to the total loss. As a result, **minority groups experience higher loss during testing, which limits the model to represent them accurately.** These factors prompt us to reduce the impact of PA imbalance on the generation of $\mathcal{S}$.

## 5 FairDD

**Overview**   In this paper, we propose a novel FDD framework that achieves both PA fairness and TA accuracy for the model trained on its generation $\mathcal{S}$, regardless of whether the original datasets exhibit PA fairness. As illustrated in Fig. 2, FairDD first partitions the dataset into different groups w.r.t. PA and then introduces an effective synchronized matching to equally align $\mathcal{S}$ with each group within $\mathcal{T}$. Compared to vanilla DDs, which pull the synthetic dataset toward the majority group in the synthetic dataset, FairDD proposes a group-level synchronized alignment, in which each group attracts the synthetic data toward itself, thus forcing it to move farther from other groups. This synchronized pull prevents the generation from collapsing into majority groups (fairness) and ensures class-level distributional coverage (accuracy).

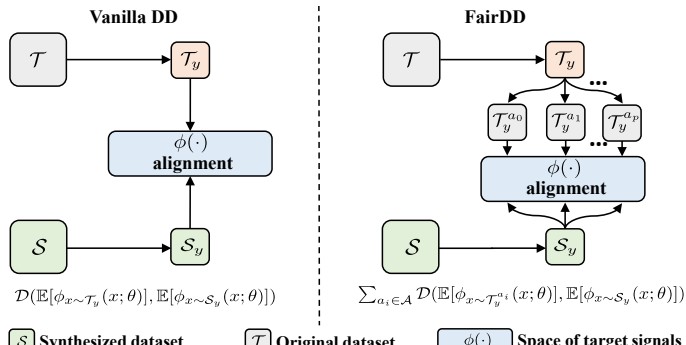

Figure 2: The overview of FairDD. FairDD first groups target signals of $\mathcal{T}$ and then proposes to align $\mathcal{S}$ (random initialization) with respective group centers. With this synchronized matching, $\mathcal{S}$ is simultaneously pulled by all group centers in a batch. This prevents the condensed dataset $\mathcal{S}$ from being biased towards the majority group, allowing it to better cover the distribution of $\mathcal{T}$.

**Synchronized matching**   As mentioned in Sec. 4, vanilla DDs fail to mitigate PA imbalance and even amplify the discrimination. The relation behind the failure is that the majority group dominates the generation direction of $\mathcal{S}$ and leads to the resulting $\mathcal{S}$ inheriting the PA imbalance, i.e., preference to fitting to the majority group. To avoid the synthetic samples collapsing into the majority group, we decompose the single target (dominated by the majority group) into PA-wise sub-targets, and simultaneously align $\mathcal{S}$ with these sub-targets, without incorporating the specific sample ratio of each group into the optimization objective. The samples assigned to one group have the same PA within the same class label. In this way, we obtain the unified objective function of FairDD:

$$\mathcal{L}_{FairDD}(\mathcal{S}; \theta, \mathcal{T}) := \sum_{y \in \mathcal{Y}} \sum_{a_i \in \mathcal{A}} \mathcal{D}(\mathbb{E}[\phi_{x \sim \mathcal{T}_y^{a_i}}(x; \theta)], \mathbb{E}[\phi_{x \sim \mathcal{S}_y}(x; \theta)]). \tag{4}$$

The reformulation forms synchronized matching, where different sub-targets attempt to pull $\mathcal{S}$ into their corresponding PA regions. Each PA group holds equal importance in generating $\mathcal{S}$, ultimately converging to a balanced (fair) status. Subsequently, we present a theoretical analysis illustrating how FairDD effectively mitigates PA imbalance and aligns TA distribution.

**Theorem 5.1.** *For any PA set $\mathcal{A}$, network parameters $\theta$, and target signs $\phi(\cdot)$, $\mathcal{L}_{FairDD}(\mathcal{S}; \theta, \mathcal{T})$ could mitigate the influence of PA imbalance of original datasets on generating synthetic samples. Especially when $\mathcal{D}(\cdot)$ is MAE or MSE, synchronized matching ensures that the signal expectation of $\mathcal{S}$ is situated at the center of the expectation across all PA groups within $\mathcal{T}$.*

*Proof.* We assume that $\mathbb{E}[\phi_{x \sim \mathcal{S}_y}(x; \theta)]$ could reach the optimal solution for each class. Hence, we have $\partial \mathcal{L}_{FairDD}(\mathcal{S}_y; \theta, \mathcal{T}_y)/\partial \mathbb{E}[\phi_{x \sim \mathcal{S}_y}(x; \theta)] = 0$:

$$\mathbb{E}[\phi_{x \sim \mathcal{S}_y}(x; \theta)] = \frac{\lambda}{p} \sum_{a_i \in \mathcal{A}} \mathbb{E}[\phi_{x \sim \mathcal{T}_y^{a_i}}(x; \theta)] \tag{5}$$

According to Eq. 5, the resulting $\mathbb{E}[\phi_{x \sim \mathcal{S}_y}(x; \theta)]$ are independent on the sample ratio $\mathcal{R}_y$, indicating the corresponding $\mathcal{S}$ unaffected by $\mathcal{R}_y$. As a result, the condensed $\mathcal{S}$ will not be dominated by majority groups that happened in vanilla DDs. All PA centers contribute equally to the generation of $\mathcal{S}$, which succeeds in mitigating the PA imbalance of $\mathcal{T}$. Especially when $\mathcal{D}(\cdot)$ is MAE or MSE, the expectation of target signs of $\mathcal{S}$ is equal to the arithmetic mean of centers of all PA groups. This shows that $\mathcal{S}$ generated by FairDD is not biased towards any groups.

Although we mitigate the bias inheritance in vanilla DDs by synchronously aligning $\mathcal{S}$ to fine-grained PA-wise groups, it is also crucial to investigate whether $\mathcal{L}_{FairDD}(\mathcal{S}; \theta, \mathcal{T})$ (synchronized matching) ensures that the resulting $\mathcal{S}$ achieves comprehensive distributional coverage for $\mathcal{T}$. As mentioned above, $\mathcal{L}(\mathcal{S}; \theta, \mathcal{T})$ matches $\mathcal{S}$ and $\mathcal{T}$ in a global view to fully cover $\mathcal{T}$'s distribution. Below, we provide a theoretical guarantee that $\mathcal{L}_{FairDD}(\mathcal{S}; \theta, \mathcal{T})$ could maintain comprehensive coverage compared to $\mathcal{L}(\mathcal{S}; \theta, \mathcal{T})$ when $\mathcal{D}(\cdot, \cdot)$ is a convex distance function, commonly used in diverse DDs [3].

**Theorem 5.2.** *For any PA set $\mathcal{A}$ and target signs $\phi_\theta(\cdot)$, $\mathcal{L}_{FairDD}(\mathcal{S}; \theta, \mathcal{T})$ is the upper bound of vanilla unified objective $\mathcal{L}(\mathcal{S}; \theta, \mathcal{T})$, i.e., $\mathcal{L}_{FairDD}(\mathcal{S}; \theta, \mathcal{T}) \geq \mathcal{L}(\mathcal{S}; \theta, \mathcal{T})$, when $\mathcal{D}(\cdot, \cdot)$ is convex. Optimizing $\mathcal{L}_{FairDD}(\mathcal{S}; \theta, \mathcal{T})$ can guarantee the comprehensive distribution coverage for $\mathcal{T}$.*

The proof is given in Appendix D. $\mathcal{L}_{FairDD}(\mathcal{S}; \theta, \mathcal{T})$ serves as the upper bound of $\mathcal{L}(\mathcal{S}; \theta, \mathcal{T})$, meaning that minimizing $\mathcal{L}_{FairDD}(\mathcal{S}; \theta, \mathcal{T})$ ensures the minimization of $\mathcal{L}(\mathcal{S}; \theta, \mathcal{T})$. Hence, optimizing $\mathcal{S}$ in FairDD can guarantee the distributional coverage by bounding $\mathcal{L}(\mathcal{S}; \theta, \mathcal{T})$ tailored for accuracy.

# 6 Experiment

## 6.1 Experiment Setup

**Datasets** Comprehensive experiments are conducted on publicly available datasets with diverse types of bias, including foreground bias (FG), background bias (BG), combined BG & FG bias, and real-world bias. The evaluated datasets include synthetic datasets: C-MNIST (FG), C-MNIST (BG), Colored-FMNIST (FG), Colored-FMNIST (BG), and CIFAR10-S (BG & FG), as well as real-world datasets: CelebA, UTKFace, and BFFHQ. For more details on these datasets, please refer to Appendix B and C. We also explore Tiny-ImageNet-S and ImageNet Subsets-S with the same operations as those performed on CIFAR10 for CIFAR10-S.

**Baselines & Evaluation metrics** FairDD is a general fairness framework applicable to diverse DDs in DMF. We apply FairDD to diverse DMF approaches including DM method DM [69] and GM methods DC [70], IDC (DC version) [27], and DREAM (DC version) [36]. To provide an overall evaluation for model bias toward PA, we use $\text{DEO}_\text{M}(\downarrow) \in [0, 100]$ and $\text{DEO}_\text{A}(\downarrow) \in [0, 100]$ to measure the worst and average fairness levels. Also, we report accuracy($\uparrow$) to assess the model's prediction of TA. We also provide a comparison with MTT in Appendix R. Sometimes, we will abuse DM+FairDD and FairDD for clarification.

**Implementation details** We default to BR of 0.9 for all synthetic original datasets to induce significant PA skew. In Table 16, we conduct the ablation study on BR. All baselines are reproduced using official implementations. FairDD doesn't introduce extra hyperparameters or learnable parameters. Experiments are conducted on PyTorch 2.0.0 with a single NVIDIA RTX 3090 24GB GPU.

## 6.2 Main results

We use distilled datasets $\mathcal{S}$ from different DDs to train and evaluate ConvNet with the same parameters, and then report the corresponding fairness and accuracy. *Random* refers to sampling defined IPC from the original dataset to create smaller datasets. Besides, *Whole* means we train the model using the entire training dataset without distillation or sampling.

**FairDD significantly improves the fairness of vanilla DDs** We provide comprehensive fairness comparisons across various DDs, including DM and DC. As illustrated in Table 1, vanilla DDs fail to mitigate the bias present in the original datasets and even exacerbate unfairness towards biased groups. In C-MNIST (FG), the distilled datasets from DM suffer from severe unfairness at IPC=10 compared

---

[3]Emperical experiments show FairDD also can cover the TA distributions when $\mathcal{D}(\cdot, \cdot)$ is not convex.

Table 1: Fairness comparison on diverse IPCs.

| Methods Dataset | IPC | Random | | DM | | DM+FairDD | | DC | | DC+FairDD | | IDC | | IDC+FairDD | | DREAM | | +FairDD | | Whole | |
|---|---|---|---|---|---|---|---|---|---|---|---|---|---|---|---|---|---|---|---|---|---|
| | | DEO$_M$ | DEO$_A$ | DEO$_M$ | DEO$_A$ | DEO$_M$ | DEO$_A$ | DEO$_M$ | DEO$_A$ | DEO$_M$ | DEO$_A$ | DEO$_M$ | DEO$_A$ | DEO$_M$ | DEO$_A$ | DEO$_M$ | DEO$_A$ | DEO$_M$ | DEO$_A$ | DEO$_M$ | DEO$_A$ |
| C-MNIST (FG) | 10 | 100.0 | 98.72 | 100.0 | 99.96 | **17.04** | **7.95** | 99.85 | 65.61 | **26.75** | **11.96** | 100.0 | 91.45 | **12.24** | **6.64** | 98.99 | 78.71 | **11.88** | **7.21** | 10.10 | 5.89 |
| | 50 | 100.0 | 99.58 | 100.0 | 91.68 | **10.05** | **5.46** | 46.99 | 20.55 | **18.42** | **8.86** | 65.34 | 34.91 | **9.18** | **5.94** | 52.03 | 26.63 | **18.37** | **7.50** | | |
| | 100 | 100.0 | 88.64 | 99.36 | 66.38 | **8.17** | **4.86** | 45.27 | 17.45 | **22.32** | **9.49** | 64.36 | 35.82 | **11.88** | **6.21** | 69.30 | 33.30 | **11.88** | **6.88** | | |
| C-MNIST (BG) | 10 | 100.0 | 99.11 | 100.0 | 99.97 | **13.42** | **6.77** | 100.0 | 73.60 | **20.66** | **9.94** | 100.0 | 88.30 | **18.61** | **7.50** | 100.0 | 52.06 | **15.31** | **6.83** | 9.70 | 5.78 |
| | 50 | 100.0 | 99.77 | 100.0 | 97.85 | **8.98** | **5.25** | 60.66 | 26.38 | **20.29** | **9.90** | 93.05 | 42.23 | **19.66** | **8.05** | 64.15 | 23.30 | **20.41** | **9.04** | | |
| | 100 | 100.0 | 89.07 | 100.0 | 52.23 | **6.60** | **4.31** | 62.63 | 20.87 | **32.58** | **10.40** | 63.24 | 27.79 | **12.24** | **6.32** | 44.88 | 22.86 | **16.33** | **7.80** | | |
| C-FMNIST (FG) | 10 | 100.0 | 99.18 | 100.0 | 99.05 | **26.87** | **16.38** | 99.40 | 78.96 | **46.80** | **24.01** | 100.0 | 97.27 | **32.33** | **16.80** | 100.0 | 95.17 | **42.00** | **20.87** | 79.20 | 41.72 |
| | 50 | 100.0 | 94.61 | 100.0 | 96.46 | **24.92** | **13.74** | 99.33 | 67.02 | **46.67** | **21.48** | 100.0 | 81.93 | **40.00** | **17.37** | 99.67 | 83.27 | **47.67** | **22.33** | | |
| | 100 | 100.0 | 94.85 | 100.0 | 85.11 | **23.83** | **12.75** | 99.58 | 66.45 | **56.68** | **23.07** | 100.0 | 79.10 | **48.33** | **17.43** | 97.33 | 70.10 | **74.00** | **40.40** | | |
| C-FMNIST (BG) | 10 | 100.0 | 99.40 | 100.0 | 99.68 | **33.05** | **19.72** | 100.0 | 92.91 | **61.75** | **34.88** | 100.0 | 99.40 | **42.00** | **23.80** | 100.0 | 94.70 | **36.00** | **23.50** | 91.40 | 51.68 |
| | 50 | 100.0 | 98.52 | 100.0 | 99.71 | **24.50** | **14.47** | 100.0 | 75.41 | **44.60** | **25.25** | 100.0 | 95.60 | **78.00** | **34.50** | 100.0 | 88.40 | **34.00** | **23.70** | | |
| | 100 | 100.0 | 96.05 | 100.0 | 93.88 | **21.95** | **13.33** | 99.70 | 73.38 | **52.75** | **23.48** | 100.0 | 90.70 | **77.00** | **36.00** | 100.0 | 83.90 | **40.00** | **23.20** | | |
| CIFAR10-S | 10 | 25.04 | 8.29 | 59.20 | 39.31 | **31.75** | **8.73** | 42.23 | 27.35 | **22.08** | **8.22** | 80.70 | 48.38 | **19.90** | **5.28** | 51.80 | 31.43 | **20.80** | **7.77** | 49.72 | 33.17 |
| | 50 | 57.11 | 28.89 | 75.13 | 55.70 | **18.28** | **7.35** | 71.46 | 45.81 | **34.39** | **11.21** | 92.00 | 60.56 | **29.00** | **9.10** | 56.80 | 36.19 | **14.70** | **6.53** | | |
| | 100 | 66.49 | 43.16 | 73.81 | 55.10 | **14.77** | **5.89** | 68.69 | 48.64 | **32.70** | **11.26** | 92.70 | 60.93 | **62.80** | **25.18** | 82.30 | 48.12 | **12.10** | **6.06** | | |
| CelebA | 10 | 10.48 | 9.20 | 30.01 | 28.85 | **9.37** | **5.71** | 15.48 | 14.16 | **6.64** | **5.29** | 34.85 | 34.48 | **8.36** | **4.49** | 40.75 | 36.70 | **9.20** | **5.36** | 24.85 | 24.16 |
| | 50 | 22.88 | 20.32 | 46.26 | 38.81 | **14.08** | **9.87** | 24.89 | 23.83 | **14.33** | **12.92** | 56.74 | 46.50 | **22.57** | **15.15** | 43.57 | 38.53 | **23.62** | **14.29** | | |
| | 100 | 18.67 | 18.01 | 42.63 | 41.12 | **10.93** | **6.65** | 29.00 | 27.52 | **18.16** | **17.04** | 50.99 | 42.66 | **28.27** | **17.63** | 52.51 | 39.34 | **24.87** | **15.36** | | |
| UTKface | 10 | 26.00 | 18.47 | 51.40 | 34.87 | **37.40** | **21.60** | 43.80 | 26.80 | **35.00** | **20.66** | 52.00 | 30.80 | **34.40** | **23.13** | 46.00 | 29.87 | **32.40** | **24.93** | 39.00 | 24.00 |
| | 50 | 40.60 | 25.27 | 43.60 | 32.13 | **23.60** | **17.27** | 38.40 | 27.20 | **27.80** | **20.86** | 44.60 | 30.73 | **30.20** | **22.40** | 38.20 | 27.73 | **29.80** | **23.13** | | |
| | 100 | 50.00 | 30.07 | 43.60 | 34.60 | **27.20** | **20.33** | 30.60 | 23.87 | **27.20** | **18.33** | 42.80 | 31.13 | **39.00** | **23.53** | 46.60 | 32.87 | **27.40** | **21.13** | | |
| BFFHQ | 10 | 19.44 | 16.72 | 44.32 | 34.76 | **15.84** | **10.68** | 47.04 | 38.92 | **37.84** | **28.84** | 58.00 | 51.92 | **21.28** | **12.44** | 60.96 | 52.76 | **25.52** | **19.36** | 66.40 | 55.20 |
| | 50 | 37.12 | 26.84 | 60.88 | 50.56 | **19.76** | **14.96** | 59.20 | 51.24 | **52.24** | **42.48** | 70.64 | 59.28 | **13.92** | **10.88** | 62.08 | 60.04 | **31.28** | **30.00** | | |
| | 100 | 43.12 | 36.20 | 65.36 | 53.12 | **17.52** | **13.32** | 60.56 | 49.76 | **54.72** | **46.64** | 66.24 | 60.68 | **14.96** | **8.20** | 65.76 | 59.60 | **10.24** | **7.12** | | |

Table 2: Accuracy comparison on diverse IPCs.

| Methods Datasets | IPC | Random | DM | +FairDD | DC | +FairDD | IDC | +FairDD | DREAM | +FairDD | Whole |
|---|---|---|---|---|---|---|---|---|---|---|---|
| | | Acc. | Acc. | Acc. | Acc. | Acc. | Acc. | Acc. | Acc. | Acc. | Acc. |
| C-MNIST (FG) | 10 | 30.75 | 25.01 | **94.61** | 71.41 | **90.62** | 53.06 | **95.67** | 75.04 | **94.04** | 97.71 |
| | 50 | 47.38 | 56.84 | **96.58** | 90.54 | **92.68** | 88.55 | **96.77** | 91.02 | **94.59** | |
| | 100 | 67.41 | 78.04 | **96.79** | 91.64 | **93.23** | 90.39 | **97.11** | 88.87 | **95.16** | |
| C-MNIST (BG) | 10 | 27.95 | 23.40 | **94.88** | 65.91 | **90.84** | 62.09 | **94.84** | 79.81 | **93.54** | 97.80 |
| | 50 | 45.52 | 47.74 | **96.86** | 88.53 | **92.20** | 86.14 | **95.29** | 89.24 | **93.20** | |
| | 100 | 67.28 | 79.87 | **97.33** | 90.20 | **92.73** | 89.66 | **95.84** | 90.70 | **94.06** | |
| C-FMNIST (FG) | 10 | 32.80 | 33.35 | **77.09** | 60.77 | **76.01** | 44.08 | **79.66** | 49.72 | **77.24** | 82.94 |
| | 50 | 42.48 | 49.94 | **82.11** | 69.08 | **75.83** | 64.45 | **80.80** | 65.69 | **78.79** | |
| | 100 | 55.31 | 57.99 | **83.25** | 68.84 | **74.91** | 66.37 | **80.28** | 68.25 | **78.51** | |
| C-FMNIST (BG) | 10 | 24.96 | 22.26 | **71.10** | 47.32 | **68.51** | 37.59 | **72.67** | 45.30 | **71.56** | 77.97 |
| | 50 | 34.92 | 36.27 | **79.07** | 60.58 | **75.80** | 46.20 | **73.72** | 53.62 | **72.80** | |
| | 100 | 44.87 | 49.30 | **80.63** | 62.70 | **71.76** | 48.61 | **73.18** | 53.32 | **73.00** | |
| CIFAR10-S | 10 | 23.60 | 37.88 | **45.17** | 37.88 | **41.82** | 48.30 | **56.40** | 55.09 | **58.40** | 69.78 |
| | 50 | 36.46 | 45.02 | **58.84** | 41.28 | **49.26** | 47.26 | **57.84** | 57.59 | **61.85** | |
| | 100 | 39.34 | 48.11 | **61.33** | 42.73 | **51.74** | 47.27 | **56.98** | 57.14 | **62.70** | |
| CelebA | 10 | 54.51 | 61.79 | **64.37** | 57.19 | **57.63** | 61.49 | **63.54** | 64.38 | **66.26** | 74.09 |
| | 50 | 55.99 | 64.61 | **68.50** | 60.16 | 59.89 | 60.75 | **66.89** | 64.62 | **68.26** | |
| | 100 | 60.62 | 65.13 | **68.84** | 62.53 | 61.89 | 64.04 | **67.24** | 62.58 | **64.12** | |
| UTKFace | 10 | 46.62 | 65.23 | **66.92** | 58.52 | **60.01** | 67.05 | **67.85** | 67.75 | 67.68 | 78.67 |
| | 50 | 59.70 | 68.94 | **71.75** | 69.00 | **70.28** | 69.82 | **69.95** | 71.97 | 71.12 | |
| | 100 | 63.87 | 71.27 | **73.70** | 66.88 | **67.65** | 72.75 | 69.43 | 70.13 | 66.42 | |
| BFFHQ | 10 | 57.40 | 64.90 | **65.46** | 62.62 | **63.30** | 65.52 | **68.70** | 64.32 | 63.94 | 71.40 |
| | 50 | 61.78 | 65.28 | **69.00** | 64.62 | **68.04** | 70.64 | 70.50 | 63.04 | **66.60** | |
| | 100 | 62.94 | 66.20 | **73.74** | 67.40 | **68.72** | 63.16 | **70.50** | 62.74 | **63.64** | |

Table 3: Cross-arch. comparison.

| Method | Cross arch. | DM | | | DM+FairDD | | |
|---|---|---|---|---|---|---|---|
| | | DEO$_M$ | DEO$_A$ | Acc. | DEO$_M$ | DEO$_A$ | Acc. |
| C-MNIST (FG) | ConvNet | 100.0 | 91.68 | 56.84 | **10.05** | **5.46** | **96.58** |
| | AlexNet | 100.0 | 98.82 | 44.02 | **10.35** | **6.16** | **96.12** |
| | VGG11 | 99.70 | 70.73 | 75.22 | **9.55** | **5.39** | **96.80** |
| | ResNet18 | 100.0 | 96.00 | 52.05 | **8.40** | **4.63** | **97.13** |
| | Mean | 99.93 | 89.31 | 57.03 | **9.59** | **5.41** | **96.66** |
| C-FMNIST (BG) | ConvNet | 100.0 | 99.71 | 36.27 | **24.50** | **14.47** | **79.07** |
| | AlexNet | 100.0 | 99.75 | 22.72 | **20.60** | **14.11** | **76.14** |
| | VGG11 | 99.70 | 97.77 | 43.11 | **21.60** | **14.36** | **78.57** |
| | ResNet18 | 100.0 | 99.78 | 23.37 | **22.50** | **14.96** | **75.21** |
| | Mean | 100.0 | 99.25 | 31.37 | **22.30** | **14.73** | **77.25** |
| CIFAR10-S | ConvNet | 75.13 | 55.70 | 45.02 | **18.28** | **7.35** | **58.84** |
| | AlexNet | 75.30 | 52.57 | 36.09 | **15.84** | **5.12** | **49.16** |
| | VGG11 | 61.48 | 44.05 | 43.23 | **11.51** | **4.16** | **52.65** |
| | ResNet18 | 76.23 | 54.35 | 38.03 | **16.44** | **5.14** | **50.93** |
| | Mean | 72.04 | 51.67 | 40.59 | **15.27** | **5.44** | **52.90** |
| CelebA | ConvNet | 40.26 | 38.81 | 64.61 | **14.08** | **9.87** | **68.50** |
| | AlexNet | 32.51 | 31.62 | 63.10 | **9.38** | **5.75** | **64.24** |
| | VGG11 | 26.03 | 24.63 | 61.57 | **8.95** | **6.32** | **62.05** |
| | ResNet18 | 25.60 | 24.93 | 60.32 | **6.72** | **4.29** | **61.80** |
| | Mean | 31.10 | 30.25 | 62.40 | **9.78** | **6.58** | **64.15** |
| UTKface | ConvNet | 43.60 | 32.13 | 68.94 | **23.60** | **17.27** | **71.75** |
| | AlexNet | 48.40 | 31.93 | 66.37 | **33.40** | **21.90** | **69.26** |
| | VGG11 | 48.20 | 30.67 | 65.93 | **32.70** | **21.03** | **67.24** |
| | ResNet18 | 50.50 | 31.77 | 62.63 | **34.70** | **20.17** | **66.79** |
| | Mean | 47.63 | 31.63 | 65.97 | **31.10** | **20.09** | **68.76** |
| BFFHQ | ConvNet | 60.88 | 50.56 | 65.28 | **19.76** | **14.96** | **69.00** |
| | AlexNet | 55.96 | 45.56 | 65.80 | **17.60** | **12.98** | **68.71** |
| | VGG11 | 57.12 | 42.88 | 66.12 | **25.16** | **16.22** | **67.79** |
| | ResNet18 | 56.88 | 46.88 | 62.60 | **23.12** | **14.14** | 63.47 |
| | Mean | 57.71 | 46.47 | 64.95 | **21.41** | **14.58** | **67.24** |

to *Whole*, with DEO$_M$ and DEO$_A$ reaching 100.0 and 99.96 vs. 10.10 and 5.89. In some cases, *Random* presents better fairness than vanilla DDs, particularly when dealing with complex objects like CelebA. This suggests that while vanilla DDs effectively condense information into smaller samples, their inductive bias, which favors the majority group, worsens the fairness to the minority group. However, when FairDD is applied to vanilla DDs, there is a significant improvement in fairness performance, with DEO$_M$ dropping substantially from 100.0 to 17.04, and DEO$_A$ decreasing from 99.96 to 7.95 in C-MNIST (FG). This indicates that FairDD's synchronized matching ensures the equal treatment of each group, effectively mitigating the bias that vanilla DDs exacerbate. FairDD further reduces the bias originally present in the original datasets. For example, DC + FairDD outperforms *Whole* in C-FMNIST (FG) and CIFAR10-S, as well as in the real-world dataset CelebA, achieving the overall improvement on DEO$_M$ and DEO$_A$ metrics. Similar performance gains are also observed in other baselines.

**FairDD maintains the comparable and even higher accuracy than vanilla DDs** A fairness framework must maintain TA accuracy in addition to improving fairness across PA groups. We report the TA accuracy of FairDD in comparison to other baselines in Table 2. Compared to *Random*, training the model by vanilla DDs yields better performance. This shows that vanilla DDs capture the informative patterns of majority groups, improving their TA accuracy. However, by focusing on dominant patterns in majority groups, they neglect the important patterns in minority groups within the training datasets. Thus, their representation coverage is limited. In contrast, FairDD proposes synchronized matching to push the $\mathcal{S}$ to cover each group, and as a result, the generated $\mathcal{S}$ retains key patterns of all groups and achieves comprehensive coverage. For example, DM obtains 25.01 at IPC = 10 on C-MNIST (FG), and its accuracy boosts to 94.61 when applying FairDD. In real-world CelebA, FairDD obtains comparable performance for DC and presents superiority over vanilla DDs. These demonstrate that FairDD could mitigate the bias without compromising accuracy.

**Generalization to diverse architectures** Here, we investigate the cross-model generalization of FairDD, where ConvNet is used to condense datasets, and we evaluate $\mathcal{S}$ on other architectures, including AlexNet, VGG11, and ResNet18. We compare DM and FairDD across four datasets at IPC = 50, evaluating performance against BG, FG, BG & FG, and real-world biases. As shown in Table 3, among these architectures, FairDD achieves $DEO_M$ of 10.05, 10.35, 9.55, and 8.40 on C-MNIST (FG), $DEO_A$ of 14.47, 14.11, 14.36, and 14.96 on C-FMNIST (BG), and accuracy of 58.84, 49.16, 52.65, and 50.93 on CIFAR10-S. These steady results suggest that $\mathcal{S}$ generated by FairDD is not restricted to the model used for distillation but generalizes well across diverse architectures. Additionally, with the model capacity increasing, the model generally tends to be more fair to all groups. However, the accuracy sometimes decreases, such as when it drops from 58.84 (ConvNet) to 50.93 (ResNet18) in CIFAR10-S and from 68.50 (ConvNet) to 61.80 (ResNet18) in CelebA. We assume that while increased attention from larger models can lead to accuracy gains for minority groups, it may limit the representations for majority groups at certain levels. The accuracy gains for minority groups may be smaller than the accuracy losses for majority groups, particularly in larger models that have limited potential improvement in recognizing minority groups.

**Discussion** FairDD employs a bias-free alignment objective, where the expectation of the distilled data is unbiased across all groups, as shown in Eq. 5. To ensure fairness during optimization, each PA group contributes equally to the total loss, with gradients that are independent of group sample sizes. This design leads to balanced sample generation across all protected groups. Regarding TA, the loss used in vanilla DDs is designed to align with TA-wise classification accuracy. In FairDD, as proven in Thm. 5.2, the loss function serves as an upper bound on the loss of vanilla DDs. This implies that minimizing the FairDD objective ensures distributional coverage across TA. As a result, models trained on FairDD-distilled datasets achieve higher accuracy on minority PA groups, while maintaining stable performance on majority groups. The main takeaways of FairDD are summarized as follows: **Fairness-Aligned Objective:** The distilled dataset should be constructed such that its expected representation is unbiased across all attribute groups, aligning with fairness principles at the objective level; **Equal Group Contribution:** Each group should contribute equally to the overall loss, encouraging balanced optimization and preventing group-specific bias during training; **Distributional Coverage of TA:** It is essential to ensure that the distilled dataset maintains adequate distributional coverage over the TA, supporting both fairness and classification accuracy.

### 6.3 Result Analysis

**Visualization analysis on fairness and accuracy** To intuitively present the effectiveness of FairDD, we train $g_\psi$ using $\mathcal{S}$ of C-MNIST (FG) distilled by DM and FairDD, and then extract the features from the test dataset. Different colors paint these resulting features according to PA and TA, respectively. As shown in Figs. 3(a) and 3(b), features with the same PA tend to form a cluster, indicating that the model trained on DM is sensitive to PA and thus failing to guarantee fairness among all PA. In contrast with DM, the feature distributions in Fig. 3(b) exhibit nearly complete overlaps across all PA. It shows that the model trained on FairDD is agnostic to PA and does not exhibit bias towards these PA. Besides the PA fairness, we also study the feature distribution from the TA perspective. Fig. 3(c) shows that features belonging to one TA scatter and fail to provide compact representations for one class. The failure of DM can be attributed to model bias toward PA. Combined with Fig. 3(a), it can be observed that PA has a stronger influence on the feature distribution compared to TA. As a result, PA-wise representations are tightly clustered, but representations from the same TA are divided into

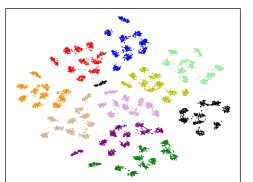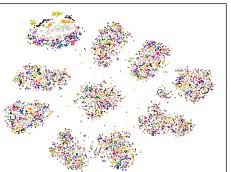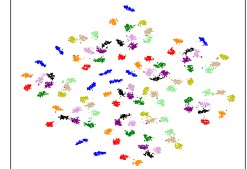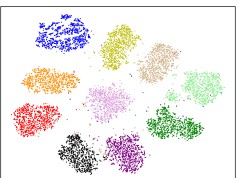

(a) PA t-SNE of DM.  (b) PA t-SNE of FairDD.  (c) TA t-SNE of DM.  (d) TA t-SNE of FairDD.

Figure 3: T-SNE visualization towards test features. Color represents distinct PA groups in (a) and (b), and TA labels in (c) and (d). In (a), DM shows obvious distinctiveness towards different PA. But (b) shows DM+FairDD eliminates the recognition of PA. In (c) and (d), DM+FairDD enables compact TA representations, but DM tends to cluster features with the same PA.

Table 5: Ablation on fair extractor.

| Methods Dataset | IPC | DM+FairDD | | | DM+LW | | | DM+LfF | | |
|---|---|---|---|---|---|---|---|---|---|---|
| | | $DEO_M$ | $DEO_A$ | Acc. | $DEO_M$ | $DEO_A$ | Acc. | $DEO_M$ | $DEO_A$ | Acc. |
| C-MNIST | 10 | **17.04** | **7.95** | **94.61** | 100.0 | 99.97 | 23.76 | 100.0 | 99.98 | 22.75 |
| (FG) | 50 | **10.05** | **5.46** | **96.58** | 100.0 | 96.64 | 48.28 | 100.0 | 91.57 | 56.23 |
| C-FMNIST | 10 | **33.05** | **19.72** | **71.10** | 100.0 | 99.66 | 21.54 | 100.0 | 99.63 | 21.74 |
| (BG) | 50 | **24.50** | **14.47** | **79.07** | 100.0 | 99.75 | 33.13 | 100.0 | 99.66 | 36.23 |
| CIFAR10-S | 10 | **31.75** | **8.73** | **45.17** | 61.21 | 42.15 | 37.26 | 77.83 | 59.22 | 43.47 |
| | 50 | **18.28** | **7.35** | **58.84** | 60.73 | 41.84 | 36.88 | 75.35 | 58.51 | 43.68 |

Table 6: Ablation on initialization at IPC = 50.

| Methods Dataset | Init. | DM | | | DM+FairDD | | |
|---|---|---|---|---|---|---|---|
| | | $DEO_M$ | $DEO_A$ | Acc. | $DEO_M$ | $DEO_A$ | Acc. |
| C-MNIST | Proportion | 100.0 | 91.68 | 56.84 | **10.05** | **5.46** | **96.58** |
| (FG) | Balanced | 100.0 | 96.41 | 53.85 | **9.80** | **5.24** | 96.48 |
| C-FMNIST | Proportion | 100.0 | 99.71 | 36.27 | **24.50** | **14.47** | 79.07 |
| (BG) | Balanced | 100.0 | 99.62 | 30.94 | **24.85** | **14.19** | **79.98** |
| CIFAR10-S | Proportion | 75.13 | 55.70 | 45.02 | **18.28** | 7.35 | **58.84** |
| | Balanced | 76.21 | 52.31 | 45.97 | **19.19** | **6.51** | 58.82 |

PA-wise parts. In contrast, FairDD proposes synchronized matching effectively mitigates this by treating each PA group equally within one TA. The equal treatment allows different PA groups within the same TA to cluster more easily, leading to more compact representations that benefit capturing class semantics in Fig. 3(d). These results highlight the superiority of FairDD in improving PA fairness and TA accuracy. Additional analysis on computation overhead and representation coverage on $\mathcal{S}$ generation are provided in Appendix J and F. We also visualize on $\mathcal{S}$ generation in Appendix G.

**Exploring the scalability on arge datasets** To further evaluate the scalability of FairDD, we conduct experiments on large datasets such as ImageNet Subset and Tiny-ImageNet. To introduce bias, we apply the same procedure used for CIFAR10-S, resulting in biased versions: ImageNet Subset-S and Tiny-ImageNet-S. The results are shown in Table 4. As observed, FairDD outperforms the vanilla DDs on these datasets, demonstrating its superior scalability.

Table 4: Scalability on ImageNet-series datasets at IPC = 10.

| Dataset | ImageNet Subset-S | | | | | | | | | | | | | | | | | | | | | Tiny-ImageNet-S | | |
|---|---|---|---|---|---|---|---|---|---|---|---|---|---|---|---|---|---|---|---|---|---|---|---|---|
| | Nette | | | Fruit | | | Woof | | | Meow | | | Squawk | | | Yellow | | | | | |
| Methods | $DEO_M$ | $DEO_A$ | Acc. | $DEO_M$ | $DEO_A$ | Acc. | $DEO_M$ | $DEO_A$ | Acc. | $DEO_M$ | $DEO_A$ | Acc. | $DEO_M$ | $DEO_A$ | Acc. | $DEO_M$ | $DEO_A$ | Acc. | $DEO_M$ | $DEO_A$ | Acc. |
| DM | 60.05 | 36.45 | 42.24 | 60.08 | 30.88 | 18.66 | 44.04 | 27.27 | 22.44 | 56.08 | 30.83 | 25.01 | 52.04 | 35.02 | 34.07 | 60.02 | 40.82 | 32.85 | 55.78 | 12.26 | 7.30 |
| DM+FairDD | **32.05** | **15.26** | **45.61** | 32.09 | 19.66 | 22.28 | 32.07 | 13.64 | 22.84 | 40.04 | 16.02 | 26.05 | 32.05 | 19.68 | 35.49 | 36.04 | 15.28 | 38.82 | 46.21 | 7.55 | 7.78 |
| DC | 46.04 | 23.28 | 40.68 | 60.84 | 29.28 | 19.56 | 45.64 | 25.48 | 21.62 | 48.04 | 23.63 | **22.92** | 47.60 | 25.12 | 31.64 | 56.06 | 30.96 | 34.44 | 54.43 | 10.33 | 9.09 |
| Dc+FairDD | **34.09** | **13.56** | **44.82** | 46.06 | 16.36 | 22.34 | 35.25 | 14.88 | 22.88 | 40.80 | 19.04 | 21.98 | 34.00 | 13.12 | 35.00 | 47.20 | 13.44 | 38.92 | 48.45 | 6.86 | 9.65 |

## 6.4 Ablation Study

**Ablation on fair extractor** General DDs treat the extractor as a non-linear transformation, where the randomly initialized extractor either does not require training or only updates parameters after a few iterations. Here, we investigate whether vanilla DDs can mitigate bias when the extractor is fair to PA. We employ two fairness approaches LW and LfF to train the extractor fairly to PA [43]. Then, we use the extractor for condensation, resulting in DM+LW and DM+LfF models. From Table 5, we can observe that DM+FairDD still outperforms DM+LW and DM+LfF on both fairness and accuracy across FG, BG, and FG&BG. Although they use fair extractor towards PA, which helps provide a balanced feature space, Eq. 3 illustrates that vanilla DDs shift synthetic datasets toward the majority group. This biased shift still causes $\mathcal{S}$ to inherit the bias of the original dataset during the condensation. In contrast, FairDD is agnostic to whether the extractor is fair and consistently mitigates the bias in the condensed dataset.

Table 7: Ablation on fairness-aware learning.

| Dataset | DC | | | DC+FairDD | | | DC+MF | | | DC+DF | | |
|---|---|---|---|---|---|---|---|---|---|---|---|---|
| | Acc. | DEO-M | DEO-A | Acc. | DEO-M | DEO-A | Acc. | DEO-M | DEO-A | Acc. | DEO-M | DEO-A |
| CMNIST-BG | 65.91 | 100.00 | 73.60 | **90.84** | **20.66** | **9.94** | 67.10 | 99.50 | 70.60 | 82.45 | 76.44 | 42.05 |
| CIFAR10-S | 37.88 | 42.23 | 27.35 | **41.82** | **22.08** | **8.22** | 39.47 | 35.86 | 22.04 | 40.10 | 27.20 | 10.29 |

**Ablation on initialization of synthetic images**   The initialization of $\mathcal{S}$ determines the prior information obtained by DDs. We examine the effect of different initialization using three strategies: *random*: randomly drawing samples from the original datasets to initialize $\mathcal{S}$; *Noise*: using noise obeying the standard normal distribution for initialization; and *balanced*: initializing with the equal number of each group. In Table 6, $\text{DEO}_M$ and $\text{DEO}_A$ metrics of DM suffer from the bias present in the original dataset across these strategies. Especially in *balanced*, we keep the synthetic dataset without group imbalance, vanilla DDs still inherit the imbalance from the original dataset. This again demonstrates the disadvantage of vanilla DDs when condensing biased datasets. In contrast, FairDD achieves robust performance in fairness and accuracy.

**Ablation on fairness-aware learning in vanilla DDs**   In this work, we explore fairness-aware learning for vanilla models through distillation fairness (DF) and model training fairness (MF). During distillation, DF assigns weights to each group inversely proportional to its sample proportion to achieve fairness-aware learning, while keeping the rest of the training procedure unchanged. MF, on the other hand, keeps the distillation process unchanged and applies fairness-aware learning only during the model training stage. We apply DF to the DC framework and obtain DC+DF, and similarly incorporate MF into DC to derive DC+MF. As illustrated in Table 7, applying these fairness regularizations can alleviate distillation bias to some extent; however, they do not ensure that the alignment objective remains unbiased across all attribute groups, nor do they guarantee comprehensive coverage of the target attribute distribution. Consequently, the distilled data of certain minority groups may be lost due to biased distillation.

**We provide a summary to guide the reader through the appendix in Appendix A. We conduct performance comparison with [13] in Appendix E and more ablation study on weighting mechanism in Appendix H, additional experiments on CelebA in Appendix I, computation overhead in Appendix J, attribute missing in test dataset in Appendix K, ablation study on the biased ratio of original datasets in Appendix L, group label noise and missing in Appendix M, balanced original dataset in Appendix N, nuanced PA groups in Appendix O, imbalanced PA groups in Appendix P, exploration on vision transformer as the backbone in Appendix Q, comparison with MTT in Appendix R.**

## 7   Conclusion

This is the first work to introduce attribute fairness into the field of dataset distillation and to systematically provide a theoretical analysis of why vanilla dataset distillation fails to mitigate attribute bias. To address the problem, we propose a unified fair dataset distillation framework called FairDD, broadly applicable to various DDs in DMF. FairDD requires no modifications to the architectures of vanilla DDs and introduces an easy-to-implement yet effective attribute-wise matching. This method mitigates the dominance of the majority group and ensures that synthetic datasets equally incorporate representative patterns with all protected attributes from both majority groups and minority groups. By doing so, FairDD guarantees the fairness of synthetic datasets while maintaining their representativeness for image recognition. We provide extensive theoretical analysis and empirical results to demonstrate the superiority of FairDD.

**Limitations**   Since FairDD relies on PA's prior information to conduct attribute-wise matching, it is valuable to explore the scenario where PA is unavailable [34]. A potential solution is to generate pseudo-labels to guide FairDD through self-supervised learning or unsupervised learning.

**Broader Impacts**   This paper aims to improve data efficiency and enhance data fairness in modern machine learning, fully compliant with legal regulations. Since training a fair model from scripts with extensive data is time-consuming, our work in providing a fair condensed dataset for effective model training can have significant societal impacts. We hope our research raises attention to achieving fairness and accuracy for dataset distillation in academia and industry.

## Acknowledgments

This work was supported by NSFC 62088101 Autonomous Intelligent Unmanned Systems and NSFC U23A20326. We thank Joey Tianyi Zhou for feedback on the draft. We also thank Ninghao Liu for helpful discussion.

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

# A Appendix summary

We summarize the appendix contents as follows:

- **Dataset details:** Appendix B
- **Dataset statistics:** Appendix C
- **Proof of the theorem:** Appendix D
- **Additional performance comparison:** Appendix E
- **Visualization analysis on representation coverage:** Appendix F
- **Visualization analysis on $\mathcal{S}$ generation:** Appendix G
- **Ablation study on weighting mechanism:** Appendix H
- **Additional experiments on CelebA:** Appendix I
- **Computation overhead:** Appendix J
- **Attribute missing in test dataset:** Appendix K
- **Ablation study on biased ratio of original datasets:** Appendix L
- **Group label noise and missing labels:** Appendix M
- **Balanced original dataset:** Appendix N
- **Nuanced PA groups:** Appendix O
- **Imbalanced PA groups:** Appendix P
- **Exploration using Vision Transformer as the backbone:** Appendix Q
- **Comparison with MTT:** Appendix R
- **More visualizations:** Appendix S

# B Datasets

Comprehensive experiments have been conducted on publicly available datasets of diverse biases, including foreground bias (FG), background bias (BG), BG & FG bias, and real-world bias. C-MNIST (FG) is a variant of MNIST [29] used to evaluate model fairness, where the handwriting numbers in each class are painted with ten different colors. To correlate the TA (digital number) and PA (color) within the training dataset, each training class is predominantly associated with one color according to the same biased ratio (BR), while the remaining samples are evenly painted with the other nine colors. BR is the ratio of the majority group samples to the total samples across all groups. For the test dataset, we evenly paint the numbers for each class with ten colors to test the model bias trained on $\mathcal{S}$. C-MNIST (BG) adopts the same operation on the background and keeps the foreground unchanged. Colored-FMNIST (FG) is the modified version of Fashion-MNIST, originally aiming to classify object semantics. Like C-MNIST (FG), we color the objects for the training and test datasets. Colored-FMNIST (BG) paints the background similarly to C-MNIST (BG). CIFAR10-S (BG & FG) introduces a PA by applying grayscale or not to CIFAR10 samples. Following [61], we grayscale a portion of the training images, correlating TA and PA among different classes. For fairness evaluation, we duplicate the test images, apply grayscale to the copies, and add them to the test dataset. We also test FairDD on the real-world facial dataset CelebA, a widely used fairness dataset. We follow the common practice of treating `attractive` attribute as TA and `gender` as PA (evaluations on other attributes refer to Appendix I).

# C Datasest statistics

In this section, we provide detailed statistics for all datasets used in the manuscript for reproduction. As shown in Table 8, we present the target attribute (TA), protected attribute (PA), the sample number of the training set, the sample number of the test set, and the BR in the training set. Additionally, all test sets are balanced, with equal sample sizes across groups. We also report the condensed ratio at IPC 10, 50, and 100, which is computed by the ratio of the condensed dataset size to the training set size.

Table 8: Statistics for all datasets used in our paper.

| Datasets | TA | PA | TA number | PA number | Training set size | Test set size | BR in Training set | BR in Test set | Condensed ratio 10 | 50 | 100 |
|---|---|---|---|---|---|---|---|---|---|---|---|
| C-MNIST (FG) | Digital number | Digital color | 10 | 10 | 60000 | 10000 | 0.90 | balance | 0.17% | 0.83% | 1.67% |
| C-MNIST (BG) | Digital number | Background color | 10 | 10 | 60000 | 10000 | 0.90 | balance | 0.17% | 0.83% | 1.67% |
| C-FMNIST (FG) | Object category | Object color | 10 | 10 | 60000 | 10000 | 0.90 | balance | 0.17% | 0.83% | 1.67% |
| C-FMNIST (BG) | Object category | Background color | 10 | 10 | 60000 | 10000 | 0.90 | balance | 0.17% | 0.83% | 1.67% |
| CIFAR10-S | Object category | Grayscale or not | 10 | 2 | 50000 | 20000 | 0.90 | balance | 0.20% | 1.00% | 2.00% |
| CelebA | Attractive | Gender | 2 | 2 | 162770 | 7656 | class0: 0.62 class1: 0.77 | balance | 0.012% | 0.061% | 0.12% |
| UTKface | Age | Race | 3 | 4 | 20813 | 1200 | class0: 0.53 class1: 0.35 class2: 0.63 | balance | 0.14% | 0.72% | 1.44% |
| BFFHQ | Age | Gender | 2 | 2 | 19200 | 1000 | class0: 0.995 class1: 0.995 | balance | 0.10% | 0.52% | 1.04% |

# D  Proof of the theorem

**Theorem D.1.** *For any PA set $\mathcal{A}$ and target signs $\phi_\theta(\cdot)$, $\mathcal{L}_{FairDD}(\mathcal{S};\theta,\mathcal{T})$ is the upper bound of vanilla unified objective $\mathcal{L}(\mathcal{S};\theta,\mathcal{T})$, i.e., $\mathcal{L}_{FairDD}(\mathcal{S};\theta,\mathcal{T}) \geq \mathcal{L}(\mathcal{S};\theta,\mathcal{T})$, when $\mathcal{D}(\cdot,\cdot)$ is convex. Optimizing $\mathcal{L}_{FairDD}(\mathcal{S};\theta,\mathcal{T})$ can guarantee the comprehensive distribution coverage for $\mathcal{T}$.*

$$\text{Proof. } \mathcal{L}(\mathcal{S};\theta,\mathcal{T}) = \sum_{y\in\mathcal{Y}} \mathcal{D}\big(\mathbb{E}[\phi_{x\sim\mathcal{T}_y}(x;\theta)], \mathbb{E}[\phi_{x\sim\mathcal{S}_y}(x;\theta)]\big)$$

$$= \sum_{y\in\mathcal{Y}} \mathcal{D}\big(\sum_{a_i\in\mathcal{A}} r_y^{a_i} \mathbb{E}[\phi_{x\sim\mathcal{T}_y^{a_i}}(x;\theta)], \mathbb{E}[\phi_{x\sim\mathcal{S}_y}(x;\theta)]\big)$$

$$\leq \sum_{y\in\mathcal{Y}} \sum_{a_i\in\mathcal{A}} r_y^{a_i} \mathcal{D}\big(\mathbb{E}[\phi_{x\sim\mathcal{T}_y^{a_i}}(x;\theta)], \mathbb{E}[\phi_{x\sim\mathcal{S}_y}(x;\theta)]\big) \tag{6}$$

$$\leq \sum_{y\in\mathcal{Y}} \sum_{a_i\in\mathcal{A}} \mathcal{D}\big(\mathbb{E}[\phi_{x\sim\mathcal{T}_y^{a_i}}(x;\theta)], \mathbb{E}[\phi_{x\sim\mathcal{S}_y}(x;\theta)]\big) \tag{7}$$

$$= \mathcal{L}_{FairDD}(\mathcal{S};\theta,\mathcal{T})$$

Eq. 6 is obtained according to Jensen Inequality, and Eq. 7 is given since group ratios are smaller than one. $\mathcal{L}_{FairDD}(\mathcal{S};\theta,\mathcal{T})$ serves as the upper bound of $\mathcal{L}(\mathcal{S};\theta,\mathcal{T})$, meaning that minimizing $\mathcal{L}_{FairDD}(\mathcal{S};\theta,\mathcal{T})$ ensures the minimization of $\mathcal{L}(\mathcal{S};\theta,\mathcal{T})$. Hence, optimizing $\mathcal{S}$ in FairDD can guarantee the distributional coverage by bounding $\mathcal{L}(\mathcal{S};\theta,\mathcal{T})$ tailored for accuracy.

# E  Additional performance comparison

Table 9: Performance comparison.

| Dataset | DM Acc. | DEO-M | DEO-A | DM+FairDD Acc. | DEO-M | DEO-A | [13] Acc. | DEO-M | DEO-A |
|---|---|---|---|---|---|---|---|---|---|
| C-MNIST-FG | 25.01 | 100.00 | 99.96 | **94.61** | **17.04** | **7.95** | 90.42 | 34.77 | 18.46 |
| CIFAR10-S | 37.88 | 59.20 | 39.31 | **45.17** | **31.75** | **8.73** | 41.42 | 48.20 | 22.48 |

Since the work does not release its code, we follow the implementation details presented in their paper. We conduct experiments on CMNIST-FG and CIFAR10-S using DM at IPC=10. Their method can mitigate the distillation bias to some extent. However, this approach does not guarantee that the alignment objective is unbiased across all attribute groups, nor does it ensure adequate distribution coverage. Instead, our methods have a fairness alignment objective to facilitate unbiased data distillation; In addition, we provide a theoretical proof to guarantee the distribution coverage for TA. We summarize the three main advantages of FairDD over theirs:

- Better preservation of target attribute representations: Their approach assigns higher weights to samples in low-density regions of the data distribution. However, these samples may lie on the periphery of the data manifold and carry low information. As a result, the distilled dataset presents the patterns with low information and hinders the distillation of informative patterns. In contrast, our method explicitly aligns the centroids of each group (defined by protected and target attributes) between the distilled and original datasets. This ensures that

each group in the distilled dataset preserves representative and informative patterns, thereby maintaining the semantic integrity of the original distribution.

- Better support for protected attribute fairness: Their weighting strategy can still underrepresent minority groups if those groups are dense in the data manifold. This results in biased generation against minority group attributes. In contrast, our method is agnostic to sample density and directly aligns groups across both PA. This ensures fair representation for all attribute groups, regardless of their density, and effectively mitigates PA-related bias.

- Stronger theoretical foundations: Their work primarily relies on empirical evaluation and does not provide a theoretical explanation for why distilled datasets inherit biases from the original data, or how their method mitigates these biases. In contrast, we offer a formal theoretical analysis that explains why dataset distillation naturally inherits bias from the source data. Furthermore, we provide provable guarantees on both target attribute accuracy and protected attribute balance.

# F   Visualization analysis on representation coverage

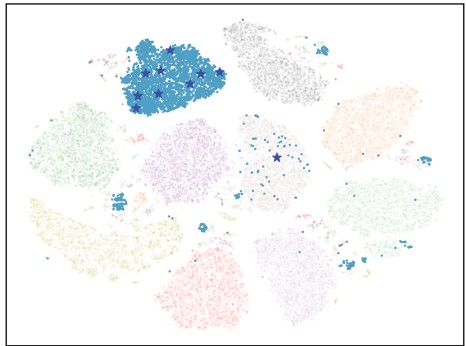 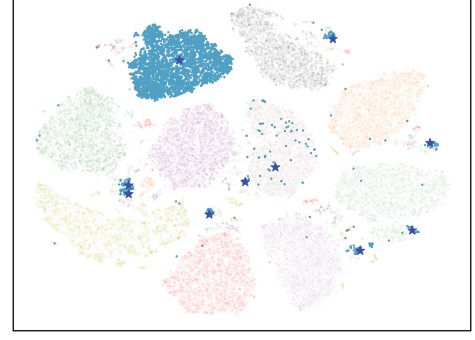

(a) The $\mathcal{S}$ distribution of generated by DM.
(b) The $\mathcal{S}$ distribution of generated by FairDD.

Figure 4: Feature coverage comparison on TA between DM and DM+FairDD. We visualize features extracted by $\phi_\theta$ on training and synthetic datasets. One class is highlighted and the remaining classes are transparent. The $\mathcal{S}$ generated by DM and FairDD are marked by stars in (a) and (b).

We investigate whether the FairDD effectively covers the whole distribution of the original datasets. For this purpose, we first feed the original training set into the randomly initialized network used in the distillation to extract the corresponding features. Subsequently, we use the same network to extract features of the distilled dataset $\mathcal{S}$ from DM and FairDD. As shown in Fig. 4(a), the synthetic samples in vanilla DDs almost locate the majority group for optimizing the original alignment objective. In this case, vanilla DDs neglect to condense the key patterns of minority groups. This leads to the information loss of minority groups in $\mathcal{S}$. FairDD achieves overall coverage for both majority and minority groups in Fig. 4(b). This is because FairDD introduces synchronized matching to reformulate the distillation objective for aligning the PA-wise groups rather than being dominated by the majority group like vanilla DDs. In doing so, FairDD avoids $\mathcal{S}$ collapsing into the majority group and retains informative patterns from all groups.

# G   Visualization analysis on $\mathcal{S}$ generation

We aim to investigate whether FaiDD renders the expectation of $\mathcal{S}$ locate the center among all groups, as clarified in Eq. 5. If the clarification holds, $\mathcal{S}$ should contain all PA at IPC = 1 because the expectation of $\mathcal{S}$ is equal to $\mathcal{S}$ when IPC =1. We visualize $\mathcal{S}$ at IPC=1 on C-MNIST (FG), where each class (digital number) is dominated by one color, and the rest is colored by the rest nine colors. As shown in Fig. 5, the $\mathcal{S}$ generated by FairDD combines all colors from PA groups. This suggests that FairDD can effectively incorporate all PA into resulting $\mathcal{S}$, indirectly validating the Theorem 5.1. Meanwhile, we observe that the majority groups dominate vanilla DDs according to Eq. 3, where the resulting $\mathcal{S}$ contains the colors from the corresponding majority groups.

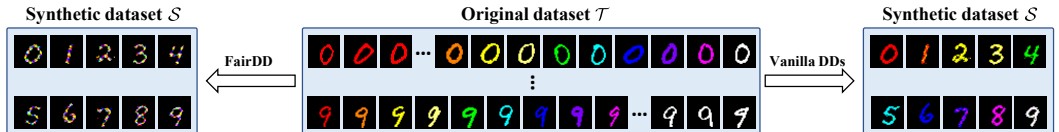

Figure 5: Visualization on $\mathcal{S}$ at IPC=1 for FairDD and vanilla DDs. **Left** is the condensed dataset using FairDD, which incorporates different PA, i.e., foreground colors. **Right** is the condensed dataset using vanilla DDs, where each class presents the same color as the corresponding majority group.

Table 10: Ablation on diverse weighting mechanisms.

| Methods Dataset | IPC | DM+FairDD | | | FairDD+IW | | | FairDD+LDAM | | | FairDD+GroupDRO | | |
|---|---|---|---|---|---|---|---|---|---|---|---|---|---|
| | | $DEO_M$ | $DEO_A$ | Acc. | $DEO_M$ | $DEO_A$ | Acc. | $DEO_M$ | $DEO_A$ | Acc. | $DEO_M$ | $DEO_A$ | Acc. |
| C-MNIST | 10 | 17.04 | **7.95** | **94.61** | **15.44** | 8.79 | 94.18 | 23.33 | 9.38 | 94.06 | 20.19 | 10.41 | 92.73 |
| (FG) | 50 | **10.05** | **5.46** | **96.58** | 12.50 | 6.49 | 96.26 | 12.03 | 6.60 | 96.32 | 17.91 | 7.90 | 94.43 |
| C-FMNIST | 10 | **26.87** | **16.38** | **71.10** | 56.60 | 35.13 | 70.55 | 64.25 | 41.13 | 69.22 | 74.50 | 42.11 | 65.19 |
| (BG) | 50 | **24.92** | **13.74** | **79.07** | 68.45 | 36.86 | 77.23 | 69.70 | 36.23 | 77.21 | 75.35 | 36.50 | 71.23 |
| CIFAR10-S | 10 | **31.75** | **8.73** | **45.17** | 48.27 | 37.41 | 38.14 | 49.88 | 36.17 | 39.27 | 44.85 | 34.53 | 38.21 |
| | 50 | **18.28** | **7.35** | **58.84** | 63.22 | 46.96 | 46.41 | 59.20 | 44.47 | 47.60 | 65.29 | 44.20 | 47.07 |

## H Ablation on weighting mechanism

Our approach treats groups separately, similar to the weighting mechanism used in the traditional fairness field. Here, we explore diverse weighting mechanisms based on our proposed group-wise alignment strategy: (1) **FairDD+IW** weights groups by inverse proportion to their respective sample size $\frac{1}{|\mathcal{T}_y^{a_i}|}$. (2) **FairDD+LDAM** adopts a soft exponential weighting $\frac{1-\beta}{1-\beta^{|\mathcal{T}_y^{a_i}|}}$ [53] (3) **FairDD+GroupDRO** optimizes the group with the maximum alignment loss instead of simultaneous alignment of all groups [49]. As illustrated in Table 10, DM + FairDD outperforms other weighting mechanisms in terms of both fairness and accuracy. We attribute the inferior performance of **FairDD+IW** and **FairDD+LDAM** to the excessive penalization of groups with larger sample sizes. Penalizing groups based on sample cardinality reintroduces an unexpected bias related to group size in the information condensation process. This results in large groups receiving smaller weights during alignment, placing them in a weaker position and causing synthetic samples to deviate excessively from large (majority) groups. Consequently, majority patterns become underrepresented, ultimately hindering overall performance. On the other hand, **FairDD+GroupDRO** shows that inadequate alignment also makes it difficult to equally represent each group. The success of FairDD lies in making each group equally contribute to the total alignment, mitigating the effects of imbalanced sample sizes across all groups. Meanwhile, FairDD performs synchronized alignment to enable the expectation of $\mathcal{S}$ to locate the expectation over all group centers of $\mathcal{T}$. Hence, FairDD can be generally applied to datasets with highly varied biases.

## I More attributes Analysis on CelebA

We explore additional facial attributes in CelebA to further demonstrate the robustness of FairDD. To this end, we regard `gender` as the PA, and `young`, `big_nose`, and `blond_hair` as the TA, which results in $\text{CelebA}_y$, $\text{CelebA}_b$, $\text{CelebA}_h$ and respectively. We also exchange the PA and TA for $\text{CelebA}_h$, resulting in $\text{CelebA}^h$ The performance is reported on fairness and accuracy in Tables 11 and 12.

## J Computation overhead

In this section, we investigate the computational efficiency of FairDD. The only computational difference between FairDD and vanilla DDs is that FairDD replaces the whole alignment with group-level alignment.

Assume we have m real samples and n synthetic samples with G attributes in a batch. For DM, the computational complexity of group-level alignment involves computing the group center. FairDD

Table 11: Fairness comparison on different attributes.

| Methods Dataset | IPC | DM DEO_M | DM DEO_A | DM+FairDD DEO_M | DM+FairDD DEO_A | DC DEO_M | DC DEO_A | DC+FairDD DEO_M | DC+FairDD DEO_A | Whole DEO_M | Whole DEO_A |
|---|---|---|---|---|---|---|---|---|---|---|---|
| CelebA$_y$ | 10 | 34.18 | 31.49 | **13.30** | **10.38** | 20.58 | 19.26 | **10.86** | **8.55** | | |
| | 50 | 46.90 | 41.13 | **12.90** | **8.21** | 27.98 | 25.18 | **14.69** | **11.26** | 25.40 | 16.02 |
| | 100 | 44.96 | 37.84 | **9.17** | **5.11** | 27.76 | 24.26 | **19.03** | **13.61** | | |
| CelebA$_b$ | 10 | 45.57 | 45.13 | **15.63** | **13.47** | 18.17 | 16.81 | **7.54** | **6.34** | | |
| | 50 | 51.91 | 51.13 | **14.44** | **12.01** | 23.85 | 22.34 | **20.58** | **16.87** | 34.48 | 25.50 |
| | 100 | 52.75 | 51.27 | **8.03** | **6.10** | 24.48 | 23.53 | **12.15** | **11.00** | | |
| CelebA$_h$ | 10 | 17.01 | 9.56 | **7.76** | **6.02** | 12.44 | 8.01 | **9.25** | **7.31** | 15.53 | 11.56 |
| CelebA$^h$ | 10 | 30.28 | 20.76 | **12.70** | **8.28** | 25.94 | 15.11 | **16.78** | **9.88** | 46.67 | 26.11 |

Table 12: Accuracy comparison.

| Methods Dataset | IPC | DM Acc. | +FairDD Acc. | DC Acc. | +FairDD Acc. | Whole Acc. |
|---|---|---|---|---|---|---|
| CelebA$_y$ | 10 | 62.34 | **63.79** | 55.91 | **56.99** | |
| | 50 | 63.59 | **67.33** | **59.87** | 59.42 | 75.99 |
| | 100 | 66.68 | **69.90** | **63.53** | 61.59 | |
| CelebA$_b$ | 10 | 57.46 | **59.50** | 52.91 | **54.67** | |
| | 50 | 58.71 | **62.39** | **56.55** | 55.46 | 66.80 |
| | 100 | 60.30 | **64.34** | **57.65** | 57.15 | |
| CelebA$_h$ | 10 | 63.64 | **64.86** | **58.04** | 57.55 | 75.33 |
| CelebA$^h$ | 10 | 77.66 | **79.71** | 72.07 | **75.03** | 79.44 |

has a complexity of G * O(m/G) + O(n). In contrast, the computational complexity of vanilla DDs is O(m) + O(n). If we ignore GPU parallelism, the computational complexity should be the same. However, since GPU parallelism is highly efficient for large batches, it results in G * O(m/G) > O(m), raising additional time consumption in FairDD. As for DC, the additional time consumption comes from two parts: one is the backward pass for gradients, and the other is to compute the average of the gradients. FairDD incurs additional memory consumption twice due to the above-mentioned GPU parallelism.

Therefore, our additional memory overhead is not related to the dataset scale but to the group number of the dataset. We evaluate the impact of the number of groups on training time (min) and peak GPU memory consumption (MB). As shown in Table 13, FairDD requires more time than vanilla DDs on C-MNIST (FG), and the time increases as the number of groups (PA) grows. This phenomenon is particularly noticeable in DC because DC suffers from GPU parallelism twice. Regarding GPU memory usage, FairDD incurs no obvious additional overhead compared to vanilla DDs.

Table 13: Comparison of computation overhead on FairDD and vanilla DDs.

| Group number | 0 (vanilla DD) T (min) | 0 (vanilla DD) G (MB) | 2 (FairDD) T (min) | 2 (FairDD) G (MB) | 4 (FairDD) T (min) | 4 (FairDD) G (MB) | 6 (FairDD) T (min) | 6 (FairDD) G (MB) | 8 (FairDD) T (min) | 8 (FairDD) G (MB) | 10 (FairDD) T (min) | 10 (FairDD) G (MB) |
|---|---|---|---|---|---|---|---|---|---|---|---|---|
| DC | 70 | 2143 | 94 | 2345 | 128 | 2369 | 152 | 2393 | 181.8 | 2419 | 210 | 2443 |
| DM | 26.2 | 1579 | 31.75 | 1579 | 33.2 | 1579 | 35.2 | 1579 | 36.5 | 1579 | 36.9 | 1579 |

Here, we further supplement the overhead analysis with respect to image resolutions. We conduct experiments on CMNIST, CelebA (32), CelebA (64), and CelebA (96) on DM and DC at IPC=10. DM and DC align different signals, which would bring different effects. As illustrated in Table 14, it can be observed that FairDD + DM does not require additional GPU memory consumption but does necessitate more time. The time gap increases from 0.42 minutes to 1.79 minutes as input resolution varies (e.g., CelebA 32 × 32, CelebA 64 × 64, and CelebA 96 × 96); however, the gap remains small. This can be attributed to FairDD performing group-level alignment on features, which is less influenced by input resolution. FairDD + DM requires no additional GPU memory consumption. Its additional time depends on both input resolutions. As for DC, FairDD requires additional GPU memory and time.

Table 14: Comparison of computation overhead for IPC = 10.

| Methods Dataset | Group number | DM Time | DM Memory | DM+FairDD Time | DM+FairDD Memory | DC Time | DC Memory | DC+FairDD Time | DC+FairDD Memory |
|---|---|---|---|---|---|---|---|---|---|
| CelebA32 × 32 | 2 | 10.93 | 2293 | 11.35 | 2293 | 32.98 | 2413 | 34.65 | 2479 |
| CelebA64 × 64 | 2 | 11.18 | 8179 | 12.20 | 8177 | 43.67 | 8525 | 47.07 | 8841 |
| CelebA96 × 96 | 2 | 12.83 | 17975 | 14.62 | 17975 | 82.37 | 18855 | 86.88 | 19437 |

## K Attribute missing in test dataset

Here, we investigate whether FairDD is agnostic to the attribute missing in the test dataset. We conduct our experiment by training the model on all PAs and testing on datasets that are missing one, two, or three PAs.

Table 15: Ablation study of missing group labels on C-MNIST and C-FMNIST under different IPC settings.

| DM | Dataset | IPC | Acc. | DEO$_M$ | DEO$_A$ |
|---|---|---|---|---|---|
| Vanilla | C-MNIST (FG) | 10 | 94.61 | 17.04 | 7.95 |
| Missing One | C-MNIST (FG) | 10 | 94.62 | 17.05 | 7.87 |
| Missing Two | C-MNIST (FG) | 10 | 94.63 | 16.79 | 7.36 |
| Missing Three | C-MNIST (FG) | 10 | 94.64 | 11.98 | 6.63 |
| Vanilla | C-MNIST (FG) | 50 | 96.58 | 10.05 | 5.46 |
| Missing One | C-MNIST (FG) | 50 | 96.60 | 10.05 | 5.38 |
| Missing Two | C-MNIST (FG) | 50 | 96.59 | 9.73 | 5.19 |
| Missing Three | C-MNIST (FG) | 50 | 96.59 | 8.53 | 4.87 |
| Vanilla | C-FMNIST (BG) | 10 | 71.10 | 33.05 | 19.72 |
| Missing One | C-FMNIST (BG) | 10 | 71.34 | 30.60 | 19.16 |
| Missing Two | C-FMNIST (BG) | 10 | 71.23 | 28.75 | 17.84 |
| Missing Three | C-FMNIST (BG) | 10 | 71.23 | 28.75 | 16.42 |
| Vanilla | C-FMNIST (BG) | 50 | 79.07 | 24.50 | 14.47 |
| Missing One | C-FMNIST (BG) | 50 | 79.31 | 23.60 | 13.72 |
| Missing Two | C-FMNIST (BG) | 50 | 79.24 | 22.90 | 13.27 |
| Missing Three | C-FMNIST (BG) | 50 | 79.24 | 22.55 | 12.58 |

Table 16: Ablation on BR at IPC = 50.

| Methods Dataset | BR | DM | | | DM+FairDD | | |
|---|---|---|---|---|---|---|---|
| | | DEO$_M$ | DEO$_A$ | Acc. | DEO$_M$ | DEO$_A$ | Acc. |
| C-MNIST (FG) | 0.85 | 99.54 | 70.13 | 76.24 | **10.13** | **5.20** | **96.62** |
| | 0.90 | 100.0 | 91.68 | 56.84 | **10.05** | **5.46** | **96.58** |
| | 0.95 | 100.0 | 100.0 | 33.73 | **10.30** | **5.84** | **96.05** |
| C-FMNIST (BG) | 0.85 | 100.0 | 95.54 | 46.14 | **23.75** | **13.85** | **79.61** |
| | 0.90 | 100.0 | 99.71 | 36.27 | **24.50** | **14.47** | **79.07** |
| | 0.95 | 100.0 | 99.79 | 26.30 | **29.15** | **17.72** | **78.46** |
| CIFAR10-S | 0.85 | 71.75 | 50.11 | 46.99 | **16.44** | **6.58** | **59.12** |
| | 0.90 | 75.13 | 55.70 | 45.02 | **18.28** | **7.35** | **58.84** |
| | 0.95 | 75.43 | 58.58 | 43.56 | **17.49** | **7.10** | **58.18** |

The missing attribute does not actually affect our performance for the following reasons. First, although the color (PA) is missing in the test dataset, its TA still contributes to the model's ability to make accurate classifications on the corresponding TA. Therefore, these missing attributes are not considered outliers in terms of TA. Second, the absence of the color in the test dataset does not impact fairness performance because FairDD is designed to generate attribute-balanced synthetic datasets. Models trained on these attribute-balanced distilled datasets are expected to treat each attribute equally. Even though the test dataset misses some existing attributes in training datasets, the model trained on such distilled datasets could still present no bias to the remaining attributes in the test dataset.

## L  Ablation on biased ratio of original datasets

BR reflects the extent of unfairness in the original datasets and indicates the level of PA skew that the distillation process of $\mathcal{S}$ will encounter. We investigate the impact of BR values on fairness performance by setting BR to $\{0.85, 0.90, 0.95\}$ on C-MNIST (FG), C-FMNIST (BG), and CIFAR10-S. The results at IPC = 50 in Table 16 show that DM is sensitive to the BR of original datasets, with its DEO$_M$ decreasing from 70.13 to 100.0 as BR increases from 0.85 to 0.95. A similar trend is observed in other datasets. Compared to DM, FairDD maintains consistent fairness and accuracy levels across different biases. This is attributed to the synchronized matching, which explicitly aligns each PA-wise subtarget, reducing sensitivity to group-specific sample numbers. This shows FairDD's robustness to PA skew in the original datasets.

## M  Ablation study on group label noise and missing

Here, we evaluate the robustness of spurious group labels could provide more insights. We randomly sample the entire dataset according to a predefined ratio. These samples are randomly assigned to

group labels to simulate noise. To ensure a thorough evaluation, we set sample ratios at 10%, 15%, 20%, and 50%. As shown in the table, when the ratio increases from 10% to 20%, the $DEO_M$ results range from 14.93% to 18.31% with no significant performance variations observed. These results indicate that FairDD is robust to noisy group labels. However, as the ratio increases further to 50%, relatively significant performance variations become apparent. It can be understood that under a high noise ratio, the excessive true samples of majority attributes are assigned to minority labels. This causes the minority group center to shift far from its true center and thus be underrepresented.

Table 17: Ablation study on group label noise.

| Methods Dataset | IPC | DM | | | DM+FairDD | | | DM+FairDD (10%) | | | DM+FairDD (15%) | | | DM+FairDD (20%) | | | DM+FairDD (50%) | | |
|---|---|---|---|---|---|---|---|---|---|---|---|---|---|---|---|---|---|---|---|
| | | Acc. | $DEO_M$ | $DEO_A$ | Acc. | $DEO_M$ | $DEO_A$ | Acc. | $DEO_M$ | $DEO_A$ | Acc. | $DEO_M$ | $DEO_A$ | Acc. | $DEO_M$ | $DEO_A$ | Acc. | $DEO_M$ | $DEO_A$ |
| CMNIST (BG) | 10 | 27.95 | 100.0 | 99.11 | 94.88 | 13.42 | 6.77 | 94.34 | 16.54 | 7.81 | 94.44 | 17.90 | 8.61 | 94.32 | 18.31 | 9.20 | 89.56 | 66.19 | 25.97 |

We investigate the experiment when the labels are missing. To provide attribute-level pseudo labels, we choose an unsupervised clustering method DBSCAN. Specifically, we do not have any group labels and use DBSCAN to cluster the samples within a batch. The clustering label is regarded as the pseudo-group label. From Table, FairDD achieves 94.77% accuracy, and 12.38% $DEO_M$ and 6.80% $DEO_A$. This demonstrates the potential of FairDD combined with an unsupervised approach when group labels are unavailable.

Table 18: Comparison of FairDD using prior vs. pseudo labels under different DDs on CMNIST-BG.

| FairDD (ipc10) | Method | Dataset | Acc | $DEO_M$ | $DEO_A$ |
|---|---|---|---|---|---|
| Prior label | FairDD + DM | CMNIST-BG | 96.86 | 13.42 | 6.77 |
| Pseudo label | FairDD + DM | CMNIST-BG | 94.77 | 12.38 | 6.80 |
| Prior label | FairDD + DC | CMNIST-BG | 90.84 | 20.66 | 9.94 |
| Pseudo label | FairDD + DC | CMNIST-BG | 90.99 | 27.96 | 10.57 |

# N Ablation study on balanced original dataset

We synthesized a fair version of CelebA, referred to as CelebA$_{Fair}$. The target attribute is attractive (attractive and unattractive), and the protected attribute is gender (female and male). In the original dataset, the sample numbers for female-attractive, female-unattractive, male-attractive, and male-unattractive groups are imbalanced. To create a fair version, CelebA$_{Fair}$ samples the number of instances based on the smallest group, ensuring equal representation across all four groups. We tested the fairness performance of FairDD and DM at IPC = 10, as well as the performance of models trained on the full dataset. As shown in Table 19, vanilla DM achieves 14.33% $DEO_A$ and 8.77% $DEO_M$. In comparison, the full dataset achieves 3.66% $DEO_A$ and 2.77% $DEO_M$. While DM still exacerbates bias with a relatively small margin, this is primarily due to partial information loss introduced during the distillation process. FairDD produces fairer results, achieving 11.11% $DEO_A$ and 6.68% $DEO_M$.

Table 19: Performance on balanced original dataset

| Methods Dataset | IPC | Whole | | | DM | | | DM+FairDD | | |
|---|---|---|---|---|---|---|---|---|---|---|
| | | Acc. | $DEO_M$ | $DEO_A$ | Acc. | $DEO_M$ | $DEO_A$ | Acc. | $DEO_M$ | $DEO_A$ |
| CelebA$_{Fair}$ | 10 | 76.33 | 3.66 | 2.77 | 63.31 | 14.33 | 8.77 | 63.17 | 11.11 | 6.68 |

# O Ablation study on nuanced PA groups

We perform a fine-grained PA division. For example, we consider gender and wearing-necktie as two correlated attributes and divide them into four groups: males with a necktie, males without a necktie, females with a necktie, and females without a necktie (CelebA$_{g\&n}$). Similarly, we consider gender and paleskin and divide them into four groups (CelebA$_{g\&p}$). Their target attribute is attractive. As shown in the Table 20, FairDD outperforms vanilla DM in the accuracy and fairness performance

on these two experiments. The performance for necktie and gender is improved from 57.50% to 25.00% on DEO$_M$ and 52.79% to 21.73% on DEO$_A$. Accuracy is also improved from 63.25% to 67.98%. Similar results can be observed for gender and paleskin. Hence, FairDD can mitigate more fine-grained attribute bias, even when there is an intersection between attributes.

Table 20: Performance on nuanced groups.

| Methods Dataset | IPC | DM | | | DM+FairDD | | |
|---|---|---|---|---|---|---|---|
| | | Acc. | DEO$_M$ | DEO$_A$ | Acc. | DEO$_M$ | DEO$_A$ |
| CelebA$_{g\&n}$ | 10 | 63.25 | 57.50 | 52.79 | 67.98 | 25.00 | 21.73 |
| CelebA$_{g\&p}$ | 10 | 62.48 | 44.81 | 41.60 | 64.37 | 26.92 | 19.33 |

## P  Ablation study on imbalanced PA groups

To further study FairDD robustness under more biased scenarios, we keep the sample number of the majority group in each class invariant and allocate the sample size to the remaining 9 minority groups with increasing ratios, i.e., 1:2:3:4:5:6:7:8:9. We denote this variant CMNIST$_{unbalance}$ This could help create varying extents of underrepresented samples for different minority groups. Notably, the least-represented PA groups account for only about 1/500 of the entire dataset, which equates to just 12 samples out of 6000 in CMNIST$_{unbalance}$. As shown in Table 21, FairDD achieves a robust performance of 16.33% DEO$_M$ and 9.01% DEO$_A$ compared to 17.04% and 7.95% in the balanced PA groups. A similar steady behavior is observed in accuracy, which changes from 94.45% to 94.61%. This illustrates the robustness of FairDD under different levels of dataset imbalance.

Table 21: Performance on imbalanced PA.

| Methods Dataset | IPC | DM | | | DM+FairDD | | |
|---|---|---|---|---|---|---|---|
| | | Acc. | DEO$_M$ | DEO$_A$ | Acc. | DEO$_M$ | DEO$_A$ |
| CMNIST | 10 | 25.01 | 100.0 | 99.96 | 94.61 | 17.04 | 7.95 |
| CMNIST$_{unbalance}$ | 10 | 23.38 | 100.0 | 99.89 | 94.45 | 16.33 | 9.01 |

## Q  Exploration on Vision Transformer as backbone

Although the Vision Transformer (ViT) is a powerful backbone network, to the best of my knowledge, current DDs, such as DM and DC, have not yet utilized ViT as the extraction network. We conducted experiments using 1-layer, 2-layer, and 3-layer ViTs. As shown in Table 22, vanilla DM at IPC=10 suffers performance degradation in classification, dropping from 25.01% to 18.63%. Moreover, as the number of layers increases, the performance deteriorates more severely. This suggests that current DDs are not directly compatible with ViTs. While FairDD still outperforms DM in both accuracy and fairness metrics, the observed improvement gain is smaller compared to results obtained on convolutional networks. Further research into leveraging ViTs for DM and FairDD is a promising direction worth exploring.

Table 22: Exploration on ViT architecture.

| Methods Dataset | IPC | DM | | | DM+FairDD | | |
|---|---|---|---|---|---|---|---|
| | | Acc. | DEO$_M$ | DEO$_A$ | Acc. | DEO$_M$ | DEO$_A$ |
| ViT1 | 10 | 18.63 | 100.0 | 98.48 | 56.15 | 82.10 | 56.72 |
| ViT2 | 10 | 18.28 | 100.0 | 98.99 | 33.89 | 72.85 | 40.97 |
| ViT3 | 10 | 16.15 | 100.0 | 95.75 | 26.70 | 65.71 | 29.46 |

# R Comparison with MTT

Unlike DMF, MTT uses a two-stage method to condense the dataset. First, it stores the model trajectories, and then it uses these trajectories to guide the generation of the synthetic dataset. To provide a comprehensive comparison, we compare FairDD with MTT, as shown in Tables 23 and 24.

Table 23: Fairness comparison on diverse IPCs. The best results are highlighted in bold.

| Methods Dataset | IPC | Random DEO$_M$ | DEO$_A$ | MTT DEO$_M$ | DEO$_A$ | DM DEO$_M$ | DEO$_A$ | DM+FairDD DEO$_M$ | DEO$_A$ | Whole DEO$_M$ | DEO$_A$ |
|---|---|---|---|---|---|---|---|---|---|---|---|
| C-MNIST (FG) | 10 | 100.0 | 98.72 | 25.70 | 14.86 | 100.0 | 99.96 | **17.04** | **7.95** | | |
| | 50 | 100.0 | 99.58 | 25.46 | 12.60 | 100.0 | 91.68 | **10.05** | **5.46** | 10.10 | 5.89 |
| | 100 | 100.0 | 88.64 | 26.81 | 13.02 | 99.36 | 66.38 | **8.17** | **4.86** | | |
| C-FMNIST (BG) | 10 | 100.0 | 99.40 | 97.00 | 62.46 | 100.0 | 99.68 | **33.05** | **19.72** | | |
| | 50 | 100.0 | 98.52 | 96.60 | 62.02 | 100.0 | 99.71 | **24.50** | **14.47** | 91.40 | 51.68 |
| | 100 | 100.0 | 96.05 | 97.20 | 63.66 | 100.0 | 93.88 | **21.95** | **13.33** | | |

Table 24: Accuracy comparison on diverse IPCs.

| Methods Dataset | IPC | Random Acc. | MTT Acc. | DM Acc. | +FairDD Acc. | Whole Acc. |
|---|---|---|---|---|---|---|
| C-MNIST (FG) | 10 | 30.75 | 92.00 | 25.01 | **94.61** | |
| | 50 | 47.38 | 94.08 | 56.84 | **96.58** | 97.71 |
| | 100 | 67.41 | 94.29 | 78.04 | **96.79** | |
| C-FMNIST (BG) | 10 | 24.96 | 67.92 | 22.26 | **71.10** | |
| | 50 | 34.92 | 70.32 | 36.27 | **79.07** | 77.97 |
| | 100 | 44.87 | 70.74 | 49.30 | **80.63** | |

From the results, FairDD outperforms MTT in both fairness and accuracy. Notably, MTT surpasses DM by a large margin, which we attribute to two factors: 1) Unlike DMF, which is directly influenced by biased data, MTT aligns the model parameters to optimize the synthetic dataset, and this indirect alignment reduces the impact of bias in the data. 2) An accurate model typically conceals its inherent unfairness, as it can better classify each class despite underlying biases. For example, when *Whole* model achieves high accuracy on the C-MNIST (FG) dataset, MTT inherits this accuracy and conceals its biases. However, when the model's accuracy declines on the C-FMNIST (BG) dataset, MTT reveals its underlying unfairness in Fig. 6(a). In contrast, FairDD directly addresses unfairness rather than relying on high accuracy to obscure biased behavior in Fig. 6(b).

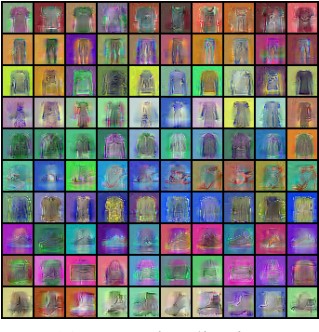
(a) MTT visualization.

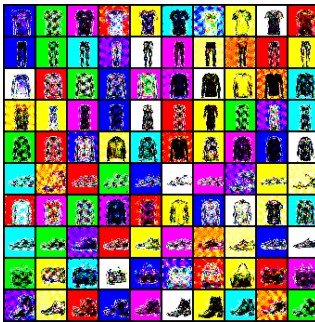
(b) FairDD visualization.

Figure 6: Visualization comparison on C-FMNIST (BG) between MTT and FairDD + DM.

# S More visualizations

We provide more visualizations at IPC = 50 on different datasets in Figures 7, 8, 9, 10, 11, and 12.

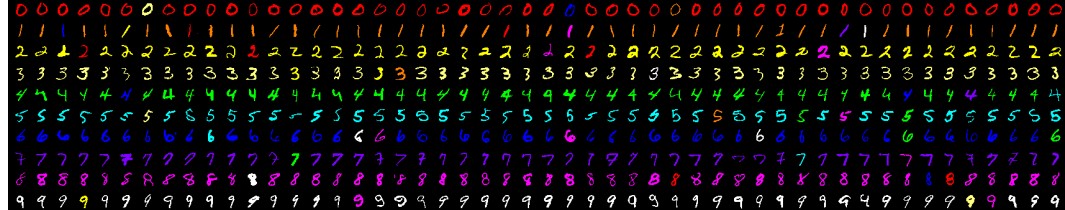

(a) Visualization of the initialized dataset at IPC = 50 in C-MNIST (FG). The foreground of each class is dominated by one color.

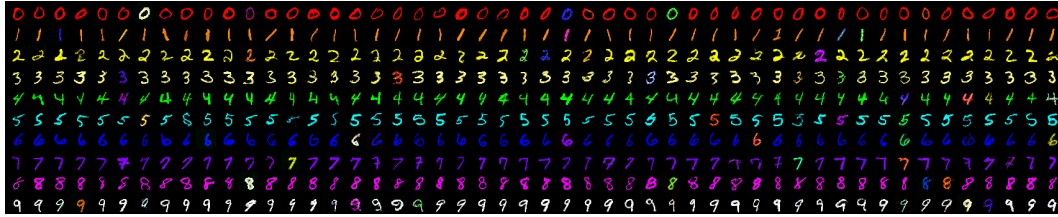

(b) Visualization of the condensed dataset at IPC = 50 in C-MNIST (FG) using Vanilla DM. The foreground of each class inherits the bias.

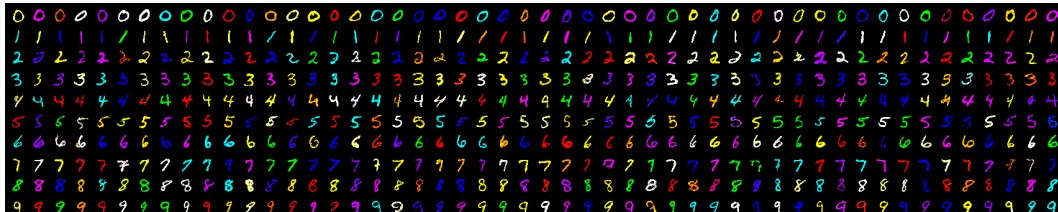

(c) Visualization of the condensed dataset at IPC = 50 in C-MNIST (FG) using FairDD + DM. The foreground of each class mitigates such bias.

Figure 7: Visualization comparison on C-MNIST (FG) between vanilla DM and FairDD + DM.

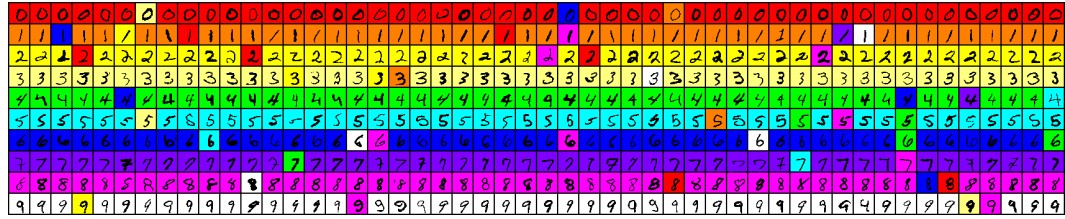

(a) Visualization of the initialized dataset at IPC = 50 in C-MNIST (BG). The background of each class is dominated by one color.

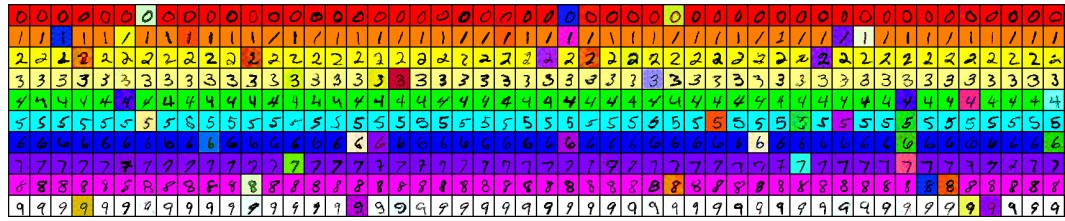

(b) Visualization of the condensed dataset at IPC = 50 in C-MNIST (BG) using Vanilla DM. The background of each class inherits the bias.

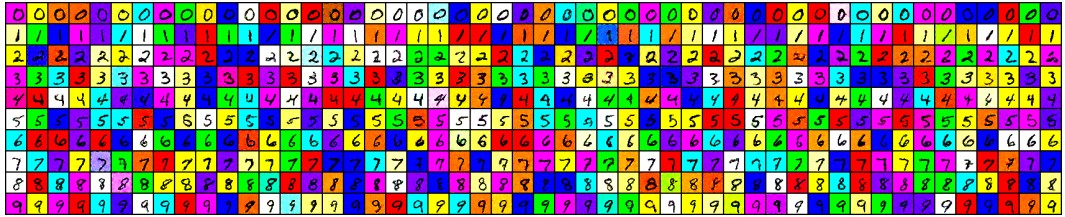

(c) Visualization of the condensed dataset at IPC = 50 in C-MNIST (BG) using FairDD + DM. The background of each class mitigates such bias.

Figure 8: Visualization comparison on C-MNIST (BG) between vanilla DM and FairDD + DM.

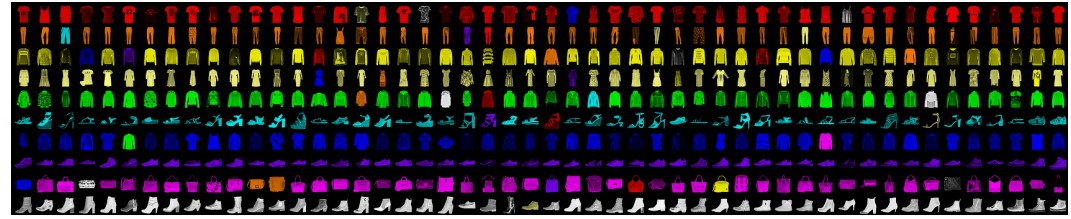

(a) Visualization of the initialized dataset at IPC = 50 in C-FMNIST (FG). The foreground of each class is dominated by one color.

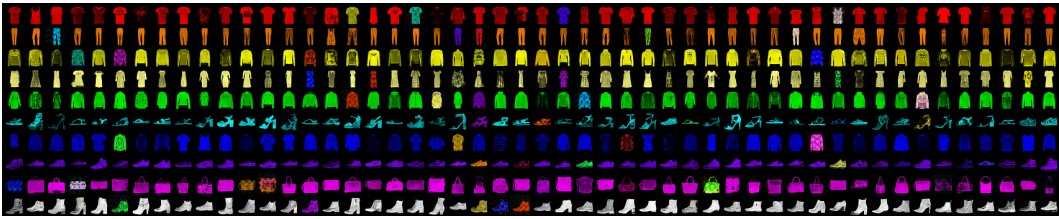

(b) Visualization of the condensed dataset at IPC = 50 in C-FMNIST (FG) using Vanilla DM. The foreground of each class inherits the bias.

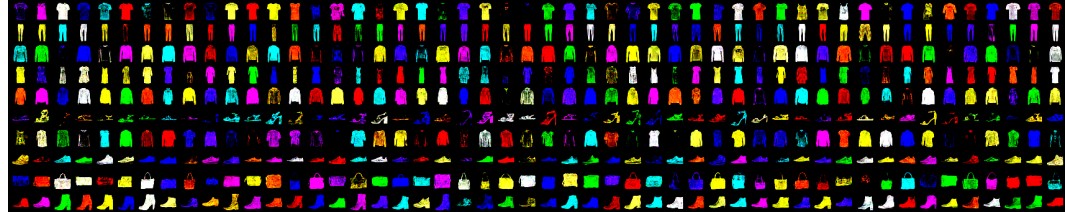

(c) Visualization of the condensed dataset at IPC = 50 in C-FMNIST (FG) using FairDD + DM. The foreground of each class mitigates such bias.

Figure 9: Visualization comparison on C-FMNIST (FG) between vanilla DM and FairDD + DM.

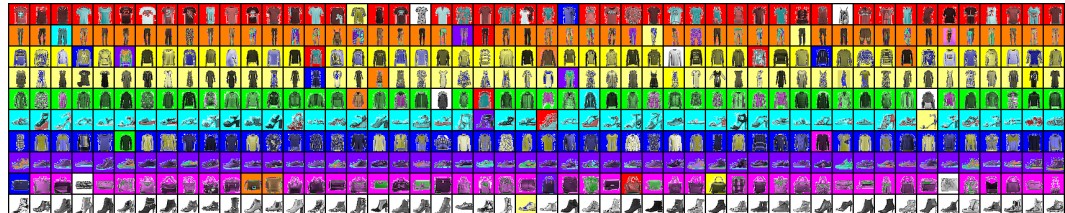

(a) Visualization of the initialized dataset at IPC = 50 in C-FMNIST (BG). The background of each class is dominated by one color.

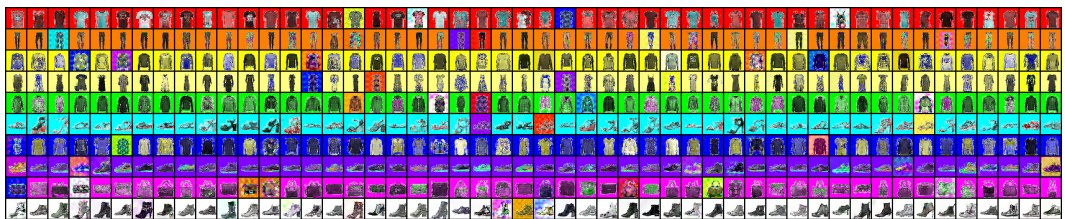

(b) Visualization of the condensed dataset at IPC = 50 in C-FMNIST (BG) using Vanilla DM. The background of each class inherits the bias.

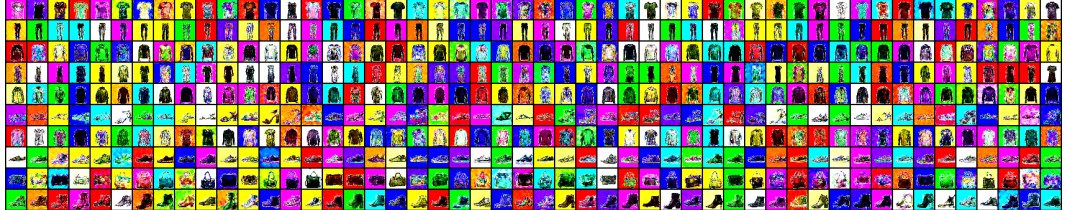

(c) Visualization of the condensed dataset at IPC = 50 in C-FMNIST (BG) using FairDD + DM. The background of each class mitigates such bias.

Figure 10: Visualization comparison on C-FMNIST (BG) between vanilla DM and FairDD + DM.

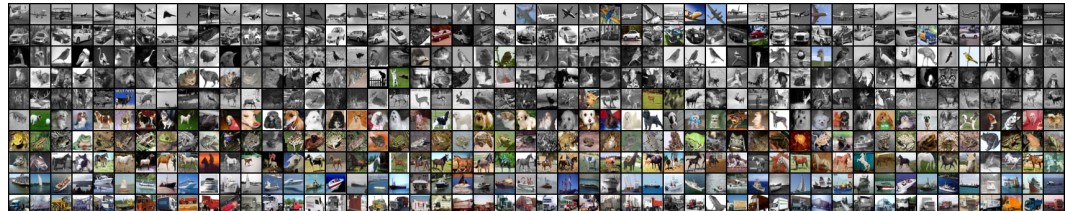

(a) Visualization of the initialized dataset at IPC = 50 in CIFAR10-S. The top five classes (rows) are dominated by the grayscale images, and color ones dominate the bottle five.

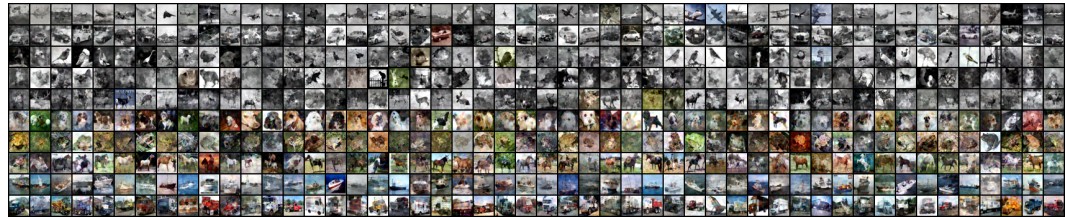

(b) Visualization of the condensed dataset at IPC = 50 in CIFAR10-S using Vanilla DM. The foreground and background of each class inherit the bias.

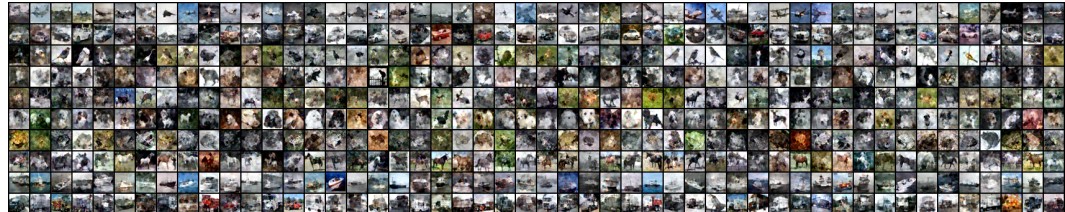

(c) Visualization of the condensed dataset at IPC = 50 in CIFAR10-S using FairDD + DM. The foreground and background of each class mitigate such bias.

Figure 11: Visualization comparison on CIFAR10-S between vanilla DM and FairDD + DM.

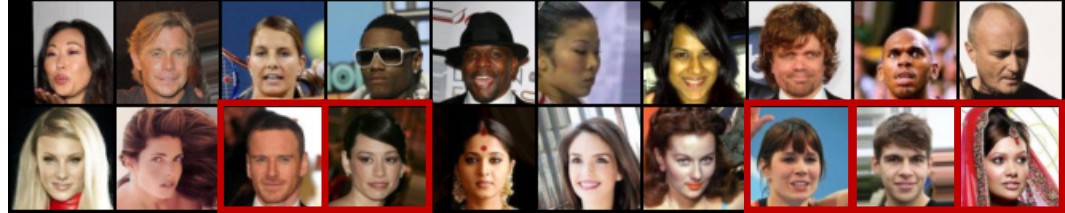

(a) Visualization of the initialized dataset at IPC = 10 in CelebA. The top row is dominated by the male, and the female dominates the bottom row.

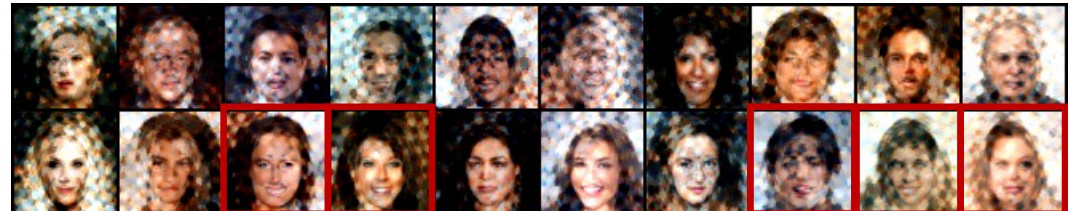

(b) Visualization of the condensed dataset at IPC = 10 in CelebA using Vanilla DM. The synthetic dataset inherits the gender bias.

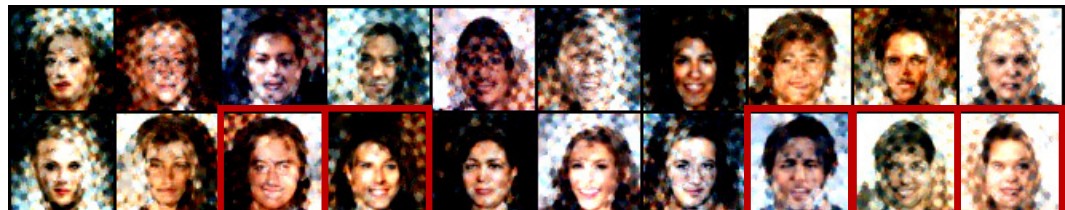

(c) Visualization of the condensed dataset at IPC = 10 in CelebA using FairDD + DM. The synthetic dataset mitigates the gender bias.

Figure 12: Visualization comparison on CelebA between vanilla DM and FairDD + DM.

