# OpenReview forum: "FairDD: Fair Dataset Distillation"
_NeurIPS.cc/2025/Conference — NeurIPS 2025 poster_

### Official Review · Reviewer_KqGz · 2025-06-10

**Clarity:** 3
**Significance:** 3
**Originality:** 3
**Rating:** 4
**Confidence:** 4

**Summary:**

This paper proposes a fair dataset distillation (DD) framework called FairDD, which matches synthetic datasets to protected attribute (PA)-wise groups of original datasets. The matching prevents collapsing into majority groups and balances generation more evenly across PA groups. Experiments show that FairDD significantly improves the fairness of vanilla DD methods.

**Questions:**

Please see the weaknesses above.

**Ethical Concerns:**

["NO or VERY MINOR ethics concerns only"]

**Final Justification:**

I carefully read the rebuttal and appreciate the clarifications on the comparison with related work, data-centric goals, sensitive attribute availability, and fairness measures along with new experiments. Assuming all these points will be reflected in the revision, I am willing to increase my score to 4.

**Limitations:**

Yes

**Quality:**

3

**Strengths And Weaknesses:**

Strengths

* Improving fairness in data distillation is an important problem.
* The paper is overall a smooth read.

Weaknesses

* The novelty of this work is questionable as there is a very similar work that has already been published at ICML 2024:
**Cui et al., "Mitigating bias in dataset distillation," ICML 2024**. The authors need to cite this work, which uses sample reweighting scheme utilizing kernel density estimation, and clearly explain and demonstrate why FairDD outperforms it. The above approach may be more advanced or at least comparable to the proposed alignment approach. Currently the baselines are just adding FairDD to existing DMF approaches, which is not sufficient.

* This work is not the first work to point out that vanilla DDs fail to mitigate bias as claimed in the Conclusion. In addition to the ICML 2024 work above, here is an earlier work whose ArXiv version is cited in the paper: **Lu et al., "Exploring the Impact of Dataset Bias on Dataset Distillation," CVPRW 2024**.
The authors should tone down their contributions.

* How do you know the right ratio without looking at the final model's fairness? The paper emphasizes reflecting the sample ratio of each group, but does not consider the learnability of groups at all. For example, even if two groups A and B have a ratio of 2:1, but B is more challenging to learn, it may be better to weight them 1:3 instead of 1:2 in order to maximize fairness.

* Another practical concern is whether the sensitive attributes are always available in image datasets. Unlike structured datasets, it is rarer for images to have this information. There needs to be a discussion what to do when this information is not available.

* If sensitive attributes are available, there are actually more fairness measures (in addition to Visual Fairness) to consider including demographic parity and equal opportunity, which should also be supported in the paper.

---

> ### Author Rebuttal · Authors · 2025-07-30
>
> **Q1: Difference from IMCL work**
>
> Thank you for bringing this work [1] to our attention. While both studies aim to address bias, they differ significantly in both scope and technical approach.
>
> **Different Scope:**
>
> Their work focuses on mitigating bias to improve classification accuracy on the TA without considering protected attributes (PA), staying within the traditional setting of DD.
>
> In contrast, our work tackles a broader challenge, which ensures fairness with respect to PA while preserving TA accuracy. We use formal evidence to illustrate that vanilla DD fails to mitigate the bias of the original dataset but amplifies the bias, and further provide an effective solution to mitigate this phenomenon in the distilled dataset. We are the first attempt to consider attribute fairness into DD, opening up a novel and underexplored direction that connects fairness and efficient data disrtillation
>
> **Different insight:**
>
> Our paper provides three main insights into fairness distillation for subsequent research:
>
> 1. Fairness-Aligned Objective: The distilled dataset should be constructed such that its expected representation is unbiased across all attribute groups, aligning with fairness principles at the data level.
>
> 2. Equal Group Contribution: Each group should contribute equally to the overall loss, encouraging balanced optimization and preventing group-specific bias during training.
>
> 3. Distributional Coverage of TA: It is essential to ensure that the distilled dataset maintains adequate distributional coverage over the target attribute (TA), supporting both fairness and classification accuracy.
>
> **Different Approach:**
>
> Their method mitigates sample-wise bias by assigning higher weights to samples located in low-density regions of the original data distribution, primarily for improving TA classification.
>
> By contrast, our method adopts a group-level alignment strategy to promote fairness across protected attribute groups. This achieves a balance between fairness and accuracy.
>
> [1] Cui, Justin, et al. "Mitigating bias in dataset distillation." In ICML 2024.
>
> **Q2: Performance comparison**
>
> Since their work does not release their code, we follow the implementation details presented in their paper. We conduct experiments on CMNIST-FG and CIFAR10-S using DM at IPC=10.
>
> |CMNIST-FG|ACC |DEO-M|DEO-A|
> | --- | --- | --- | --- |
> |DM | 25.01| 100.0| 99.96 |
> |DM +FairDD|94.61 | 17.04 |7.95|
> |ICML|90.42 | 34.77 |18.46|
>
> |CIFAR10-S| ACC |DEO-M|DEO-A|
> | --- | --- | --- | --- |
> |DM | 37.88| 59.20| 39.31 |
> |DM +FairDD| 45.17 | 31.75 |8.73|
> |ICML| 41.42 |48.20|22.48|
>
> Their method can mitigate the distillation bias to some extent. However, this approach does not guarantee that the alignment objective is unbiased across all attribute groups, nor does it ensure adequate distribution coverage. Instead, our methods have a fairness alignment objective to facilitate unbiased data distillation; In addition, we provide a theoretical proof to guarantee the distribution coverage for TA.
>
> We summarize the three main advantages of FairDD over theirs:
>
> (1) Better preservation of target attribute representations:
> Their approach assigns higher weights to samples in low-density regions of the data distribution. However, these samples may lie on the periphery of the data manifold and carry low information. As a result, the distilled dataset presents the patterns with low information and hinders the distillation of informative patterns. In contrast, our method explicitly aligns the centroids of each group (defined by protected and target attributes) between the distilled and original datasets. This ensures that each group in the distilled dataset preserves representative and informative patterns, thereby maintaining the semantic integrity of the original distribution.
>
> (2) Better support for protected attribute fairness:
> Their weighting strategy can still underrepresent minority groups if those groups are dense in the data manifold. This results in biased generation against minority group attributes. In contrast, our method is agnostic to sample density and directly aligns groups across both PA. This ensures fair representation for all attribute groups, regardless of their density, and effectively mitigates PA-related bias.
>
> (3) Stronger theoretical foundations:
> Their work primarily relies on empirical evaluation and does not provide a theoretical explanation for why distilled datasets inherit biases from the original data, or how their method mitigates these biases. In contrast, we offer a formal theoretical analysis that explains why dataset distillation naturally inherits bias from the source data. Furthermore, we provide provable guarantees on both target attribute accuracy and protected attribute balance.
>
> Nonetheless, this does not undermine the significance of their method as a pioneering contribution toward bias mitigation in classification tasks.
>
> We will supplement the related discussion in the revised version.
>
> **Q3: Comparison with CVPRW work**
>
> Thank you for pointing this out. The referenced work [2] investigates the impact of bias on TA accuracy primarily from an empirical perspective. In contrast, our work is, to the best of our knowledge, the first to provide a theoretical explanation for why vanilla DD fails to mitigate bias. Moreover, we introduce attribute fairness into the DD framework and broaden the scope of dataset distillation to consider both performance and fairness. Our method is grounded in a theoretical foundation, offering formal guarantees for both TA accuracy and PA fairness. **We will revise the Conclusion to more accurately reflect these distinctions and acknowledge prior work accordingly:** "`This is the first work to introduce attribute fairness into the field of dataset distillation and to systematically provide a theoretical analysis of why vanilla dataset distillation fails to mitigate attribute bias.`"
>
> [2] Lu, Yao, et al. "Exploring the impact of dataset bias on dataset distillation." In CVPRW 2024.
>
> **Q4: How do you know the right ratio without looking at the final model's fairness?**
>
> Thank you for your insightful comments. We would like to clarify that our goal is to mitigate attribute-wise bias present in the original dataset in a manner that is agnostic to the downstream model. The idea that an attribute-balanced dataset can help reduce bias is a well-established concept in the deep learning community. Building on this foundation, our method achieves model fairness without relying on additional regularization techniques.
>
> For your point about the learnability of each attribute, **this is fundamentally a model-centric problem, as it depends on the characteristics of the downstream model. In contrast, both DD and our proposed FairDD are data-centric tasks, agnostic to any specific model architecture.** While incorporating model-specific information into data generation may be a valid direction, **it aligns more closely with the domain of model-level fairness: how to train a fair model given a dataset. In contrast, FairDD addresses data-level fairness: how to generate a distilled dataset that is inherently unbiased, agnostic to the downstream model.** These two perspectives approach fairness from distinct yet complementary directions.
>
> **Q5: Another practical concern is whether the sensitive attributes are always available in image datasets.**
>
> As is common in many research fields, progress often begins with supervised approaches before transitioning to unsupervised solutions. For example, the representative method GroupDRO [3] also needs the group label to achieve fairness. After that, there are many approaches without labels. The need for labels does not diminish their contribution to model fairness. We believe this is a necessary process that calls for community-wide efforts rather than individual solutions.
>
> We investigate the case where such labels are unavailable. As detailed in Appendix L, we apply the unsupervised clustering algorithm DBSCAN to generate pseudo attribute labels, and show in Table 16 that FairDD maintains good performance even under this setting. This indicates that our method is compatible with label-free scenarios, at least to some degree.
>
> || CMNIST-BG |ACC |DEO-M|DEO-A|
> | --- | --- | --- | --- | --- |
> |DC|Prior label| 90.84| 20.66| 9.94 |
> |DC|Pseudo label|90.99 | 27.96 |10.57|
>
> [3] Sagawa, S., Koh, P. W., Hashimoto, T. B., & Liang, P. (2020). "Distributionally Robust Neural Networks for Group Shifts: On the Importance of Regularization for Worst-Case Generalization." In ICLR 2020.
>
> **Q6: Including demographic parity and equal opportunity, which should also be supported in the paper.**
>
> Equal Opportunity (EO) means that all groups should have the same true positive rate. DEO measures how much this condition is violated when there are multiple groups or attributes. DEO-M captures the worst-case difference in true positive rates between any two groups, while DEO-A computes the average difference across all groups.
>
> Demographic Parity (DP) requires that all groups receive positive predictions at the same rate, regardless of their true labels. However, in our work, we are specifically concerned with the fairness of true positive samples. Therefore, we adopt DEO-M and DEO-A as our evaluation metrics, both of which are widely used in prior research. While it is always possible to compute additional metrics for a more comprehensive evaluation, we argue that DEO-M and DEO-A already sufficiently evaluate the fairness in our setting.

---

> > ### Comment · Reviewer_KqGz · 2025-08-04
> >
> > I carefully read the rebuttal and appreciate the clarifications on the comparison with related work, data-centric goals, sensitive attribute availability, and fairness measures along with new experiments. Assuming all these points will be reflected in the revision, I am willing to increase my score.

---

> > > ### Author Response · Authors · 2025-08-05
> > >
> > > We are glad to hear that your concerns have been addressed. We are willing to incorporate your valuable comments and make the following updates to our manuscript:
> > >
> > > 1. Thorough discussion and comparison with previous work, including papers from ICML and CVPRW.
> > > 2. Clearer emphasis on the data-centric goals of our approach.
> > > 3. In-depth analysis of the availability of sensitive attributes.
> > > 4. More detailed illustration and explanation of the fairness measures used.

---

### Official Review · Reviewer_cwZj · 2025-06-20

**Clarity:** 3
**Significance:** 3
**Originality:** 3
**Rating:** 4
**Confidence:** 3

**Summary:**

The paper introduces a method to address fairness issues in the topic of dataset distillation (DD). Specifically, when distilling image instances in DD, the recognition performance for minor catetories / attributes drops, amplifying unfairness. Therefore, in this paper, by organizing protected attributes (PAs) separately and then processing & performing DD, the paper realizes a method called FairDD for the purpose of fairness treatment.

**Questions:**

- In the proposed FairDD, does treating fairness (protect PAs) generally tend to improve accuracy as well? By protecting minor yet important image feature representations, does FairDD reduce the visually similar features in the distillated image dataset, this means that it is acting as a training regularization? Moreover, if we consider other factors beyond fairness, DD may be further enhanced by the preserved and improved image features and enhanced overall performance?

- The reviewer believes that [Nakamura et al., ECCV 2024] is closely related work. The research work uses a "1p-frac dataset" generated by fractal geometry to successfully pre-train a Vision Transformer (ViT) from a single synthetic sample. In the appendix, the present paper mentions that applying ViT did not work very well at this moment. The insights from their fractal-based approach for ViT pre-training help the next approach for using ViT as a backbone in the topic of FairDD / DD? Also, clarifying the relationship between DD and such that approaches, formula-driven supervised learning (FDSL), might further advance the DD topic.

[Nakamura et al., ECCV 2024] R. Nakamura et al. “Scaling Backwards: Minimal Synthetic Pre-training?”, ECCV 2024.

**Ethical Concerns:**

["NO or VERY MINOR ethics concerns only"]

**Final Justification:**

The reviewer is grateful to the authors for great efforts and discussion in the rebuttal phase. In the rebuttal term, the reviewer has had a better understanding on the trial regarding the fairness issues on dataset distillation. This is enough to possess an accept-side paper rating. Hopefully the FairDD and their thoughtful knowledge can be widely shared in the research community. Moreover, the reviewer is looking forward to seeing the next step towards the improved FairDD in the near future.

**Limitations:**

Yes

**Paper Formatting Concerns:**

No concerns about paper formatting.

**Quality:**

3

**Strengths And Weaknesses:**

- S1. The FairDD paper tackles the problem of unfairness in DD. The conventional DD distillates a labeled image dataset into a smaller synthetic dataset for training, however, the distillated image dataset tends to eliminate image instances which have minor categories / attributes. It tends to focus on majority categories / attributes. In order to achieve fairer image recognition in this context, this work, FairDD, preserves the truly protected instances during the process of dataset distillation, this is defined as a FairDD in this paper.

- S2. The proposed approaches and their established model used to improve both performance and fairness scores in the proposed FairDD are quite reasonable. For example, as shown in Figure 2, PA groups are split, and image instances can be separately distilled. By processing all PA groups simultaneously, FairDD avoids focusing only on major categories / attributes and ensures that minority categories / attributes are also represented at that time. Theoretical guarantees are further shown that the FairDD removes imbalance and improves fairness inside of the distillated datasets.

- S3. In the experimental results against baseline / vanilla DD methods, the proposed FairDD successfully improves fairness metrics (DEO_M / DEO_A; lower is better) while also enhancing recognition accuracy (higher is better). The proposed method, FairDD, is evaluated comprehensively on representative datasets, namely C-MNIST, C-FMNIST, CIFAR10-S, CelebA, UTKFace, BFFHQ as shown in Tables 1, 2, and 3, and outperforms baselines on almost all metrics in the presented paper. Additional analyses include visualizations, scalability studies, and an ablation study examining the fair extractor and synthetic image initialization.

- W1. Although extensive experiments in the experimental section at the main paper and appendices partially clarify FairDD’s merits and limitations, the paper will gain greater value by including a more comprehensive discussion. Specifically, a comprehensive description why FairDD works so well in terms of fairness scores and recognition performance. Here, this description should be appeared in the main paper rather than the appendices.

---

> ### Author Rebuttal · Authors · 2025-07-30
>
> Thanks for your effort in reviewing our paper.
>
> **Q1: Why FairDD works so well in terms of fairness scores and recognition performance.**
>
> Thank you for your valuable comments. The superiority of FairDD originates from its ability to achieve unbiased generation with respect to protected attributes (PA) and to provide distributional coverage guarantees for target attributes (TA).
>
> Specifically, for PA, FairDD employs a bias-free alignment objective, where the expectation of the distilled data is unbiased across all groups, as shown in Equation (5). To ensure fairness during optimization, each PA group contributes equally to the total loss, with gradients that are independent of group sample sizes. This design leads to balanced sample generation across all protected groups.
>
> Regarding TA, the loss used in vanilla DD is designed to align with TA-wise classification accuracy. In FairDD, as proven in Theorem 5.2, the loss function serves as an upper bound on the vanilla DD loss. This implies that minimizing the FairDD objective ensures distributional coverage across TA.
>
> As a result, models trained on FairDD-distilled datasets achieve higher accuracy on minority PA groups, while maintaining stable performance on majority groups. Therefore, FairDD generally achieves a more effective trade-off between fairness and accuracy compared to traditional dataset distillation approaches. It is worth noting that although accuracy on majority groups may slightly decrease, dataset distillation is inherently a data-hungry task. In this context, the accuracy gain on minority groups typically outweighs the slight reduction on majority groups
>
>
> **Q2: In the proposed FairDD, does treating fairness (protect PAs) generally tend to improve accuracy as well?**
>
> Yes, please refer to the above answers.
>
> **Q3: The insights from their fractal-based approach for ViT pre-training help the next approach for using ViT as a backbone in the topic of FairDD / DD?**
>
> Thank you for bringing this reference to our attention. This work also explores training models with limited data, sharing a similar objective with DD. We will include it in the Related Work section as a complementary approach.
>
> In methods like DM and DC, the backbone or feature extractor is used to approximate the non-linear transformation from input to representation. However, learning on the backbone during distillation can limit the generalizability of the distilled data. This is why such methods typically network with random initialization or restrict the network training to just a few optimization steps when distilling the dataset.
>
> From my perspective, the challenge of applying dataset distillation to Transformers does not stem from the ViT pre-training itself, but rather from the nature of the self-attention mechanism. Unlike CNNs, which preserve local spatial structure, self-attention in Transformers mixes information across image patches from the very first layer. This early interleaving of patch-level features makes it more difficult for distilled data to encode coherent semantic cues.
>
> **Q4: Clarifying the relationship between DD and such that approaches, formula-driven supervised learning (FDSL), might further advance the DD topic.**
>
> FDSL and DD share a common objective: enabling effective model training using a limited amount of data. However, they approach this goal from different directions. FDSL starts with a small number of procedurally generated examples and expands the data manifold through structured variations (e.g., formula-driven perturbations) to expose the model to diverse representations. In contrast, DD begins with a large dataset and compresses it into a compact, information-rich synthetic set that mimics the training effect of the original data.
>
> From a technical perspective, these approaches could be orthogonal and potentially complementary. For example, one could first apply DD to compress a large real dataset into a small synthetic one, and then apply FDSL techniques to expand the utility of that distilled dataset. This combined strategy could further improve training efficiency and generalization of DD.

---

> > ### Author Response · Authors · 2025-08-07
> >
> > As the discussion comes to a close, please let us know if you have any remaining questions that need to be addressed. We would be happy to assist further.

---

> > > ### Comment · Reviewer_cwZj · 2025-08-07
> > > **Official Comment by Reviewer cwZj**
> > >
> > > Thank you very much for the detailed response. In particular, the reviewer is interested in Q4 as it contributes to expanding the DD topic itself. At this point, the reviewer has no further concerns regarding the questions. Therefore, the paper rating is expected to remain on the acceptance side.

---

> ### Author Response · Authors · 2025-08-08
>
> We appreciate your thoughtful comments and are pleased that your questions have been resolved.
>
> FDSL could potentially be combined with label distillation as a post-distillation approach, which may further unleash the potential of the distilled dataset. This is a promising direction worth further exploration.
>
> Thank you again for helping us improve the paper.

---

### Official Review · Reviewer_yyqk · 2025-06-30

**Clarity:** 3
**Significance:** 3
**Originality:** 3
**Rating:** 5
**Confidence:** 3

**Summary:**

This paper introduces FairDD, a fairness-aware dataset distillation framework designed to reduce bias towards protected attributes in synthetic datasets, without sacrificing target attribute accuracy. By aligning distilled data to protected attribute-wise distributions rather than majority-dominated global ones, FairDD improves fairness across multiple dataset distillation paradigms and shows strong performance that is justified theoretically.

**Questions:**

None

**Ethical Concerns:**

["NO or VERY MINOR ethics concerns only"]

**Final Justification:**

N/A

**Limitations:**

See Strengths and Weaknesses field.

**Quality:**

4

**Strengths And Weaknesses:**

Strengths:
1. The paper addresses a critical yet underexplored issue, namely bias amplification in dataset distillation, and suggests FairDD, the first method to address this challenge.
2. The authors provide formal proofs that FairDD mitigates protected attribute imbalance and ensures target attribute distributional coverage.
3. Evaluation involves both synthetic and real-world datasets, showing consistent fairness improvements. Also, comprehensive ablations are reported in the appendix.

Weaknesses
1. The method requires known protected attribute labels, limiting its applicability to real-world scenarios where such labels are often unavailable and inferring pseudo-labels can be very challenging.

2. While the technical content is strong, the writing could be clearer in dense theoretical sections (e.g., Sec. 4), which currently may hinder accessibility to a broader ML audience. Specifically:
  a. (lines 125-126) The concepts of Gradient Matching and Distribution Matching have not been adequately explained neither here nor in the related works. The authors should elaborate more on how the gradient information and embedding distribution matching is formulated and achieved.
  b. (line 130) Abbreviations IDC and DREAM are not defined and citations are missing.
  c. (lines 129-130) How important is the choice of distance metric, and why are these particular metrics better suited for these methods?
 d. (lines 151-153) The fact that bias-aligned samples contribute more to the loss does not necessarily mean that the loss cannot be minimized for bias-conflicting samples. In typical classification tasks, previous work [1] has investigated the loss and gradient spaces with respect to bias-aligned and bias-conflicting samples to demonstrate how bias is introduced. Does this also apply to the Dataset Distillation task?
 e. Considering the previous comment, the authors should elaborate further on the two arguments (in bold) presented in lines 155–158.
 f. It is not clear whether the suggested method can generalize for multiple protected attributes.
g. The authors should include an algorithm in Section 5 that outlines the proposed method to enhance the reproducibility of their work.

3. While Worst Group Accuracy is a standard evaluation metric in the context of mitigating visual biases, the authors opt for Equalized Odds. It would be helpful if they could elaborate on the rationale behind this choice.

[1] Sarridis, I., Koutlis, C., Papadopoulos, S., & Diou, C. (2024). BAdd: Bias Mitigation through Bias Addition. *arXiv preprint arXiv:2408.11439*.

---

> ### Author Rebuttal · Authors · 2025-07-30
>
> Thank you for your efforts in reviewing our paper.
>
> **Q1: Prior to protected attribute labels**
>
> Although our approach requires attribute labels, we want to clarify that, as is common in many research fields, progress often begins with supervised approaches before transitioning to unsupervised solutions. For example, the representative method GroupDRO [1] also needs the group label to achieve fairness. After that, there are many approaches without labels. The need for labels does not diminish their contribution to model fairness. We believe this is a necessary process that calls for community-wide efforts rather than individual solutions.
>
> We also investigate the case where such labels are unavailable. As detailed in Appendix L, we apply the unsupervised clustering algorithm DBSCAN to generate pseudo attribute labels, and show in Table 16 that FairDD maintains good performance even under this setting. This indicates that our method is compatible with label-free scenarios, at least to some degree. We would like to leave the deep exploration for future work.
>
> || CMNIST-BG |ACC |DEO-M|DEO-A|
> | --- | --- | --- | --- | --- |
> |DC|Prior label| 90.84| 20.66| 9.94 |
> |DC|Pseudo label|90.99 | 27.96 |10.57|
>
> [1] Sagawa, S., Koh, P. W., Hashimoto, T. B., & Liang, P. (2020). "Distributionally Robust Neural Networks for Group Shifts: On the Importance of Regularization for Worst-Case Generalization." In ICLR 2020.
>
> **Q2: The detailed concepts of Gradient Matching and Distribution Matching**
>
> Distribution matching formulates the distilled process by alignment the embedding distribution between the original dataset and the distilled dataset. They minimize the MMD to update the distilled data.
> $
> \mathcal{L}(\mathcal{S}; \theta, \mathcal{T}) :=\sum\_{y \in \mathcal{Y}} ||\mathbb{E}\_{x \sim \mathcal{T}\_y} \left[ \phi(x; \theta) \right]-\mathbb{E}\_{x \sim \mathcal{S}\_y} \left[ \phi(x; \theta) \right]||^2 $
>
> Then, we can rewrite it into the attribute-level formulation:
>
> $
> \mathcal{L}(\mathcal{S}; \theta, \mathcal{T}) :=\sum_{y \in \mathcal{Y}} ||\sum_{a_i \in \mathcal{A}} r_y^{a_i} \cdot \mathbb{E}\_{x \sim \mathcal{T}_y^{a_i}} \left[ \phi(x; \theta) \right] - \mathbb{E}\_{x \sim \mathcal{S}\_y} \left[ \phi(x; \theta) \right]||^2 \newline
> $
>
> In gradient matching methods, the gradients of the classification loss on the original and distilled datasets are aligned using either the mean absolute error (MAE) or cosine distance.
>
> $\mathcal{L}(\mathcal{S}; \theta, \mathcal{T}) :=\sum\_{y \in \mathcal{Y}} |\mathbb{E}\_{x \sim \mathcal{T}\_y} \left[\nabla\_\theta \mathcal{L}\^c(x; \theta) \right]-\mathbb{E}\_{x \sim \mathcal{S}\_y} \left[ \nabla\_\theta \mathcal{L}\^c(x; \theta) \right]|$ or $\mathcal{L}(\mathcal{S}; \theta, \mathcal{T}) :=\sum\_{y \in \mathcal{Y}} \text{Cosine-distance}(\mathbb{E}\_{x \sim \mathcal{T}\_y} \left[\nabla\_\theta \mathcal{L}\^c(x; \theta) \right], \mathbb{E}\_{x \sim \mathcal{S}\_y} \left[ \nabla\_\theta \mathcal{L}\^c(x; \theta) \right])$
>
> Then, we can rewrite it into the attribute-level formulation:
> $\mathcal{L}(\mathcal{S}; \theta, \mathcal{T}) :=\sum\_{y \in \mathcal{Y}} |\sum_{a_i \in \mathcal{A}} r_y^{a_i} \cdot \mathbb{E}\_{x \sim \mathcal{T}_y^{a_i}}\left[\nabla\_\theta \mathcal{L}\^c(x; \theta) \right]-\mathbb{E}\_{x \sim \mathcal{S}\_y} \left[ \nabla\_\theta \mathcal{L}\^c(x; \theta) \right]|$ or
> $\mathcal{L}(\mathcal{S}; \theta, \mathcal{T}) :=\sum\_{y \in \mathcal{Y}} \text{Cosine-distance}(\sum\_{a\_i \in \mathcal{A}} r\_y^{a\_i} \cdot \mathbb{E}\_{x \sim \mathcal{T}\_y^{a_i}}\left[\nabla\_\theta \mathcal{L}\^c(x; \theta) \right], \mathbb{E}\_{x \sim \mathcal{S}\_y} \left[ \nabla\_\theta \mathcal{L}\^c(x; \theta) \right])$
>
> We will supplement the related illustration in the revised version.
>
>
> **Q3: Abbreviations IDC and DREAM are not defined and citations are missing**
>
> Thank you for your detailed feedback. IDC and DREAM are two improved variants of DMF. IDC parameterizes the distilled data, while DREAM optimizes the initialization of the distilled data. However, both methods do not alter the optimization objective or the gradient flow. We will include additional illustrations and proper citations in the revised version.
>
> **Q4: How important is the choice of distance metric, and why are these particular metrics better suited for these methods?**
>
> We use the same distance metric as in their paper to ensure consistency and fair comparison. The distance choice could be found in their own work. Our method is generalizable and can be applied with various distance metrics.
>
> **Q5: Does the previous work BAdd also apply to the Dataset Distillation task?**
>
> In biased datasets, vanilla models rely on bias-aligned samples to reduce loss, but this prevents them from learning fair representations. BAdd solves this by explicitly adding bias features, allowing the model to keep the loss low on bias-aligned samples without encoding the bias itself. This frees the main representation to focus on fair, unbiased features. From my perspective, BAdd leverages the higher loss of bias-aligned samples, compared to bias-conflicting samples, to guide training more efficiently. This could potentially promote equal optimization across different groups. However, the fairness of the distilled dataset originates from
>
> 1) no biased alignment objective between the original data and distilled data (alignment objective) in Equation 5;
>
> 2)  each protected attribute should receive balanced optimization (agnostic to the sample number within attributes) during training.
>
> Therefore, Badd provides a potential solution for solution 2 but needs a deep exploration to solve 1.
>
> **Q6: Considering the previous comment, the authors should elaborate further on the two arguments (in bold) presented in lines 155–158.**
>
> The first argument (in bold) is not related to gradient behavior. It illustrates that the alignment objective itself must be unbiased across all groups. Only when the learning objective is fair and the optimization process is balanced across different attributes, can the distilled dataset be fair.
>
> For the second argument, we further elaborate that **minority groups often receive suboptimal optimization compared to their majority groups in vanilla models**. This imbalance limits the model’s ability to accurately represent its representations. This phenomenon is supported by studies such as Distributionally Robust Optimization (DRO) and Group DRO, which observe that:
>
> "A model that learns this spurious correlation would be accurate on average on an i.i.d. test set but suffer high error on groups of data where the correlation does not hold."
>
>
> **Q7: It is not clear whether the suggested method can generalize for multiple protected attributes.**
>
> Thank you for your insightful comments.
>
> FairDD can naturally handle instances with multiple protected attributes. In such cases, we assign each sample to a fine-grained group, where all samples share the same combination of attribute values. For example, if the dataset contains two attributes A = {a₁, a₂} and B = {b₁, b₂}, this results in four distinct groups: {a₁, b₁}, {a₁, b₂}, {a₂, b₁}, and {a₂, b₂}. As shown in Equation (5), each group contributes equally to the generation of the distilled dataset.
>
> We conduct experiments on CelebA under two multi-attribute settings at IPC=10, Gender + Wearing Necktie and Gender + Pale Skin
>
> The results, shown in Table 18 (Appendix N), demonstrate that FairDD remains effective in preserving fairness while maintaining competitive accuracy under multi-attribute scenarios:
>
> |CelebA| Gender + Necktie |ACC |DEO-M|DEO-A|
> | --- | --- | --- | --- | --- |
> |DM || 63.25| 57.50| 52.79 |
> |DM +FairDD| |67.98 | 25.00 |21.73|
>
> |CelebA| Gender + Pale Skin |ACC |DEO-M|DEO-A|
> | --- | --- | --- | --- | --- |
> |DM || 62.48| 44.81| 41.60 |
> |DM +FairDD| |64.37 | 26.92 |19.33|
>
> **Q8: The authors should include an algorithm in Section 5 that outlines the proposed method to enhance the reproducibility of their work.**
>
> We will supplement it in the revised version.
>
> **Q9: While Worst Group Accuracy is a standard evaluation metric in the context of mitigating visual biases, the authors opt for Equalized Odds. It would be helpful if they could elaborate on the rationale behind this choice.**
>
>  Worst Group Accuracy (WGA) is a valuable metric for evaluating fairness, as it captures the performance on the most disadvantaged group. However, it fails to provide fine-grained fairness evaluation across all groups. For example, WGA remains unchanged regardless of whether discrepancies among the remaining groups are small or large. Instead, Visual Fairness and metrics such as DEO-M and DEO-A reflect fine-grained discrepancies across multiple attributes. This allows us to measure how each group or attribute is treated in the model's predictions, providing a more localized and attribute-level fairness assessment. Therefore, we consider WGA a global fairness metric, while DEO-M and DEO-A serve as local, attribute-specific metrics. Since our goal is to generate balanced samples across all protected attributes in the distilled dataset, DEO-M and DEO-A are particularly well-suited for evaluating generation bias in each group.

---

> > ### Author Response · Authors · 2025-08-07
> >
> > As the discussion comes to a close, please let us know if you have any remaining questions that need to be addressed. We would be happy to assist further.

---

### Official Review · Reviewer_GpUd · 2025-07-03

**Clarity:** 3
**Significance:** 3
**Originality:** 2
**Rating:** 4
**Confidence:** 3

**Summary:**

This work argues that existing dataset distillation methodologies are biased towards majority protected attribute groups and fail to achieve unfairness towards minority protected attribute groups with theoretical evidences. To solve this problem, this work proposes the FairDD framework, which divides a given original dataset according to attributes and aligns each sub-dataset with a learnable small dataset. The authors conduct extensive experiments on a wide range of benchmark datasets and experimentally demonstrate that FairDD simultaneously improves both accuracy and fairness on an unbiased test dataset.

**Questions:**

Please refer to the weaknesses also.
1. I am curious about the derivation of Eqs. 3 and 5.
2. Is FairDD applicable to datasets where each instance has more than one attribute? I suspect it would be difficult to apply Attribute-aware partitioning naively in this situation.

**Ethical Concerns:**

["NO or VERY MINOR ethics concerns only"]

**Final Justification:**

In my review, I raised questions about 1) the novelty of the proposed methodology, 2) additional comparative experiments under various baselines and experimental settings, and 3) the theoretical derivation. During the rebuttal period, the authors provided 1) further explanation of the paper’s contributions, 2) additional experimental results, and 3) the details of theoretical derivation.

I still believe that the proposed method, which is just partitioning the dataset to fit PA, lacks sufficient novelty. However, I consider the following contributions to be substantial: 1) the paper is the first to address fairness in the context of dataset distillation, 2) the theoretical justification for the proposed method (Theorem 5.1), and 3) the demonstrated improvements in performance. Furthermore, the additional experimental results provided during the rebuttal are also appropriate. For these reasons, I recommend a final score of 4 (Borderline Accept).

**Limitations:**

See weaknesses and questions

**Paper Formatting Concerns:**

No paper formatting concerns.

**Quality:**

3

**Strengths And Weaknesses:**

**Strengths**
1. To the best of my knowledge, this research is the first work to discuss the fairness in dataset distillation.

2. I like the analysis of why vanilla DD fails in PA imbalance situations. It is easy for the reader to follow why we should explore dataset distillation methodologies for fairness.

3. Through the extensive experimental results, FairDD has demonstrated high performance gains despite its very simple form.

**Weaknesses**
1. My major concern is the novelty of the proposed method. FairDD partitions the original dataset into several sub-datasets w.r.t PA to mitigate the domination of the majority group. However, partitioning requires that every instance of the original dataset has a labelled attribute. This may not be realistic and becomes a shortcoming that limits the scope of application of FairDD. In addition, the methodology of dividing the real dataset according to the attribute is a simple but strong baseline, but the contribution is marginal to be called a core methodology. Finally, Attribute-based partitioning has clear drawbacks that make it difficult to apply to bilevel optimization and the trajectory matching framework.

2. Lack of sufficient comparison to a baseline that could be simple but effective. For example, a methodology that applies vanilla DD and uses fairness-aware learning to train the network in the test phase could be a baseline. I also believe that gradient matching for fairness-aware loss is a baseline. Finally, this research should also include whether bias also occurs in bilevel optimization or trajectory matching using fairness-aware learning. As this research is the first to study the possibility of dataset distillation in a new context, it is necessary to make detailed comparisons with various possible baselines.

---

> ### Author Rebuttal · Authors · 2025-07-30
>
> We thank you for your efforts in reviewing our paper.
>
> **Q1: The contributions of the proposed method.**
>
> Thank you for pointing out your concerns. We use formal evidence to illustrate that vanilla DD fails to mitigate the bias of the original dataset but amplifies the bias, and further provide an effective solution to mitigate this phenomenon in the distilled dataset. We are the first attempt to consider attribute fairness into DD, opening up a novel and underexplored direction that connects fairness and efficient data disrtillation
>
> **Insights for Subsequent Research:**
>
> 1. Fairness-Aligned Objective: The distilled dataset should be constructed such that its expected representation is unbiased across all attribute groups, aligning with fairness principles at the objective level.
>
> 2. Equal Group Contribution: Each group should contribute equally to the overall loss, encouraging balanced optimization and preventing group-specific bias during training.
>
> 3. Distributional Coverage of TA: It is essential to ensure that the distilled dataset maintains adequate distributional coverage over the TA, supporting both fairness and classification accuracy.
>
>
> **Technical contribution:**
>
> While partitioning based on PA may appear straightforward, FairDD is built upon three key components that directly enhance fairness in dataset distillation:
>
> 1. Bias-Free Alignment Objective:
> FairDD introduces a group-wise alignment objective to ensure that the expected representation of each group in the distilled data remains unbiased—an essential requirement for achieving group-level fairness. This principle is formally expressed in Equation (5).
>
> 2. Equal Group Contribution to the Loss:
> The framework enforces that each group contributes equally to the total loss during optimization. This encourages balanced gradient updates and equitable learning across all attribute groups.
>
> 3. Theoretical Guarantee over Vanilla DD:
> Whereas vanilla dataset distillation focuses exclusively on optimizing classification accuracy for TA, FairDD incorporates a fairness-aware loss function. As proven in Theorem 5.2, this loss provides an upper bound on the vanilla DD loss, integrating fairness objectives without compromising overall performance.
>
> **Q2: Reliance on attribute labels.**
>
> We investigate the case where such labels are unavailable. As detailed in Appendix L, we apply the unsupervised clustering algorithm DBSCAN to generate pseudo attribute labels, and show in Table 16 that FairDD maintains good performance even under this setting. This indicates that our method is compatible with label-free scenarios, at least to some degree.
>
> || CMNIST-BG |ACC |DEO-M|DEO-A|
> | --- | --- | --- | --- | --- |
> |DC|Prior label| 90.84| 20.66| 9.94 |
> |DC|Pseudo label|90.99 | 27.96 |10.57|
>
> As is common in many research fields, progress often begins with supervised approaches before transitioning to unsupervised solutions. For example, the representative method GroupDRO also needs the group label to achieve the fairness regularization. After that, there are many approaches without labels. The need for labels does not diminish their contribution to model fairness. We believe this is a necessary process that calls for community-wide efforts rather than individual solutions.
>
> We hope these clarifications address your concerns.
>
> **Q3: More baseline**
>
> We thank you for your comments.
>
> **For fairness-aware learning at IPC=10, we assign weights to each group based on the inverse of its sample proportion.**
>
> Since the attributes of the distilled data cannot be obtained using vanilla DD, we manually annotate the attribute labels of the distilled data. These labels are then used to train the network with a fairness-aware loss.
>
> | CMNIST-BG |ACC |DEO-M|DEO-A|
> | --- | --- | --- | --- |
> |DC  | 65.91| 100.0 | 73.60 |
> |DC+FairDD | 90.84| 20.66 |  9.94 |
> |DC+ Model fairness learning   | 67.10 | 99.50 |70.60|
> |DC +Fairness loss| 82.45 | 76.44 |42.05|
>
> | CIFAR10-S|ACC |DEO-M|DEO-A|
> | --- | --- | --- | --- |
> |DC  | 37.88| 42.23 | 27.35 |
> |DC+FairDD | 41.82| 22.08 | 8.22 |
> |DC + Model fairness learning   | 39.47 | 35.86 |22.04|
> |DC + Fairness loss| 40.10 | 27.20 |10.29|
>
> The distilled data from some minority groups was lost due to biased distillation. Although we apply fairness-aware learning to train the network, the missing attributes cannot be well represented.
>
> We also apply a fairness loss to optimize the distillation process of DC, resulting in a variant referred to as DC + Fairness loss.
>
> As for MTT [1], we use fairness loss to train the teacher network, and then use the teacher trajectory to align student one.
>
> As for bi-level optimization, we use RFAD [2], a bi-level optimization approach, for evaluation. We use the weighted fairness loss to optimize the data generation
>
> | CMNIST-BG |ACC |DEO-M|DEO-A|
> | --- | --- | --- | --- |
> |MTT| 10.92| 75.83| 32.35 |
> |MTT+Fairness loss|93.58 | 29.70 |12.40|
>
> | CIFAR10-S|ACC |DEO-M|DEO-A|
> | --- | --- | --- | --- |
> |MTT| 56.92| 37.21 | 26.85 |
> |MTT+Fairness loss| 59.52 | 24.03 |12.05|
>
> | CMNIST-BG |ACC |DEO-M|DEO-A|
> | --- | --- | --- | --- |
> |RFAD| 15.84| 100.0| 98.82 |
> |RFAD+Fairness loss|14.68 | 100.0 |99.16|
>
> | CIFAR10-S|ACC |DEO-M|DEO-A|
> | --- | --- | --- | --- |
> |RFAD| 26.25| 60.54 | 42.58 |
> |RFAD+Fairness loss| 28.56 | 55.46 |35.48|
>
> Vanilla DD with fairness loss can mitigate the distillation bias to some extent. However, this approach does not guarantee that the alignment objective is unbiased across all attribute groups, nor does it ensure distributional coverage for TA.
>
> [1] Cazenavette, George, et al. "Dataset distillation by matching training trajectories." In CVPR 2022.
>
> [2] Loo, Noel, et al. "Efficient dataset distillation using random feature approximation." In NIPS 2022.
>
>
> **Q4:  Doubt about the derivation of Eqs. 3 and 5.**
>
> \begin{aligned}
> &\mathcal{L}(\mathcal{S}; \theta, \mathcal{T}) := \textstyle
> \sum_{y \in \mathcal{Y}} \mathcal{D} \sum_{a_i \in \mathcal{A}} r_y^{a_i} \cdot \mathbb{E}\_{x \sim \mathcal{T}_y^{a_i}} \left[ \phi(x; \theta) \right],\mathbb{E}\_{x \sim \mathcal{S}_y} \left[ \phi(x; \theta) \right] \newline
> & \frac{\partial \mathcal{L}}{\partial {\partial \mathbb{E}[\phi\_{x \sim \mathcal{S}\_y}(x; \theta)]}}=\textstyle\frac{\partial \mathcal{D} (\sum\_{a_i \in \mathcal{A}}r\_y^{a\_{i}} \mathbb{E}[\phi\_{x \sim \mathcal{T}\_y^{a_i}}(x;\theta)], \mathbb{E}[\phi\_{x \sim \mathcal{S}\_y}(x;\theta)])}{\partial \mathbb{E}[\phi\_{x \sim \mathcal{S}\_y}(x; \theta)]} = 0
> \end{aligned}
>
> For distribution matching, $\mathcal{D} = MSE$:
>
> $
> \textstyle\frac{\partial ||\sum\_{a_i \in \mathcal{A}}r\_y^{a\_{i}} \mathbb{E}[\phi\_{x \sim \mathcal{T}\_y^{a_i}}(x;\theta)] -  \mathbb{E}[\phi\_{x \sim \mathcal{S}\_y}(x;\theta)])||^2}{\partial \mathbb{E}[\phi\_{x \sim \mathcal{S}\_y}(x; \theta)]} = 0
> $,
>
> we have
>
> \begin{align}
> \mathbb{E}[\phi\_{x \sim \mathcal{S}\_y}(x; \theta)]=\sum\_{a\_i \in \mathcal{A}} r\_y^{a\_{i}}\mathbb{E}[\phi\_{x \sim \mathcal{T}\_y^{a\_i}}(x; \theta)],
> \end{align}
>
> For gradient matching, $\mathcal{D} = MAE$ or  $\mathcal{D} = $Cosine distance:
>
> $
> \textstyle\frac{\partial |\sum\_{a_i \in \mathcal{A}}r\_y^{a\_{i}} \mathbb{E}[\phi\_{x \sim \mathcal{T}\_y^{a_i}}(x;\theta)] -  \mathbb{E}[\phi\_{x \sim \mathcal{S}\_y}(x;\theta)])|}{\partial \mathbb{E}[\phi\_{x \sim \mathcal{S}\_y}(x; \theta)]} = 0
> $,
>
> we have
>
> \begin{align}
> \mathbb{E}[\phi\_{x \sim \mathcal{S}\_y}(x; \theta)]= \sum\_{a\_i \in \mathcal{A}} r\_y^{a\_{i}}\mathbb{E}[\phi\_{x \sim \mathcal{T}\_y^{a\_i}}(x; \theta)],
> \end{align}
>
> When $\mathcal{D} = $Cosine distance,
> $
> \textstyle\frac{\partial |\sum\_{a_i \in \mathcal{A}}r\_y^{a\_{i}} \mathbb{E}[\phi\_{x \sim \mathcal{T}\_y^{a_i}}(x;\theta)] -  \mathbb{E}[\phi\_{x \sim \mathcal{S}\_y}(x;\theta)])|}{\partial \mathbb{E}[\phi\_{x \sim \mathcal{S}\_y}(x; \theta)]} = 0
> $,
>
> we have
>
> \begin{align}
> \mathbb{E}[\phi\_{x \sim \mathcal{S}\_y}(x; \theta)]= \textstyle \frac{\Vert \mathbb{E}[\phi_{x \sim \mathcal{S}y}(x; \theta)] \Vert_2}{\Vert \sum_{a_i \in \mathcal{A}} r_y^{a_{i}}\mathbb{E}[\phi_{x \sim \mathcal{T}_y^{a_i}}(x; \theta)]\Vert_2} \sum\_{a\_i \in \mathcal{A}} r\_y^{a\_{i}}\mathbb{E}[\phi\_{x \sim \mathcal{T}\_y^{a\_i}}(x; \theta)],
> \end{align}
> Note that $\textstyle \frac{\Vert \mathbb{E}[\phi\_{x \sim \mathcal{S}y}(x; \theta)] \Vert_2}{\Vert \sum\_{a\_i \in \mathcal{A}} r\_y^{a\_{i}}\mathbb{E}[\phi\_{x \sim \mathcal{T}\_y^{a\_i}}(x; \theta)]\Vert_2}$ is a constant to the different groups. Therefore, we could derive vanilla DDs
>
> \begin{align}
> \mathbb{E}[\phi\_{x \sim \mathcal{S}\_y}(x; \theta)]=\lambda\sum\_{a\_i \in \mathcal{A}} r\_y^{a\_{i}}\mathbb{E}[\phi\_{x \sim \mathcal{T}\_y^{a\_i}}(x; \theta)], \lambda \enspace \text{is a constant}.
> \end{align}
>
> The same analysis could be applied to FairDD.
>
> **Q5: Is FairDD applicable to datasets where each instance has more than one attribute?**
>
> Thank you for your insightful comments.
>
> FairDD can naturally handle instances with multiple protected attributes. In such cases, we assign each sample to a fine-grained group, where all samples share the same combination of attribute values. For example, if the dataset contains two attributes A = {a₁, a₂} and B = {b₁, b₂}, this results in four distinct groups: {a₁, b₁}, {a₁, b₂}, {a₂, b₁}, and {a₂, b₂}. As shown in Equation (5), each group contributes equally to the generation of the distilled dataset.
>
> To evaluate this capability, we conduct experiments on CelebA under two multi-attribute settings:
>
> Gender + Wearing Necktie
>
> Gender + Pale Skin
>
> The results, shown in Table 18 (Appendix N), demonstrate that FairDD remains effective in preserving fairness while maintaining competitive accuracy under multi-attribute scenarios:
>
> |CelebA| Gender + Necktie |ACC |DEO-M|DEO-A|
> | --- | --- | --- | --- | --- |
> |DM || 63.25| 57.50| 52.79 |
> |DM +FairDD| |67.98 | 25.00 |21.73|
>
> |CelebA| Gender + Pale Skin |ACC |DEO-M|DEO-A|
> | --- | --- | --- | --- | --- |
> |DM || 62.48| 44.81| 41.60 |
> |DM +FairDD| |64.37 | 26.92 |19.33|

---

> > ### Comment · Reviewer_GpUd · 2025-08-04
> >
> > I would like to thank the authors for their responses.
> >
> > Most of my concerns have been addressed, and I will raise my score to 4.

---

> > > ### Author Response · Authors · 2025-08-04
> > >
> > > We're glad to hear that most of your concerns have been addressed. If you have any further questions, please don’t hesitate to let us know. We are more than willing to assist with any remaining issues.

---

### Decision · Program_Chairs · 2025-09-17

**Decision:**

Accept (poster)

**Comment:**

The paper introduces FairDD, a fairness-aware dataset distillation framework that partitions original data by protected attributes (PA) and aligns synthetic subsets accordingly. Reviewers agreed that the paper identifies and addresses an important and underexplored problem. They found the theoretical justification (particularly Theorem 5.1) and the comprehensive empirical evaluation across multiple datasets convincing. The results consistently demonstrate that FairDD improves fairness while also enhancing accuracy, which reviewers considered a strong point.

The reviewers raised some concerns about the novelty, applicability, and baselines. Despite these limitations, reviewers recognized the contributions of being the first to explicitly formalize fairness in dataset distillation, providing theoretical grounding, and showing strong empirical benefits. With clarifications and additional results provided in the rebuttal, the consensus is that the strengths outweigh the weaknesses.
The AC thus recommends acceptance.